# Measuring Goal-Directedness

**Matt MacDermott**
Imperial College London

**James Fox**
University of Oxford
London Initiative for Safe AI

**Francesco Belardinelli**
Imperial College London

**Tom Everitt**
Google DeepMind

## Abstract

We define *maximum entropy goal-directedness (MEG)*, a formal measure of goal-directedness in causal models and Markov decision processes, and give algorithms for computing it. Measuring goal-directedness is important, as it is a critical element of many concerns about harm from AI. It is also of philosophical interest, as goal-directedness is a key aspect of agency. MEG is based on an adaptation of the maximum causal entropy framework used in inverse reinforcement learning. It can measure goal-directedness with respect to a known utility function, a hypothesis class of utility functions, or a set of random variables. We prove that MEG satisfies several desiderata and demonstrate our algorithms with small-scale experiments [1].

## 1 Introduction

In order to build more useful AI systems, a natural inclination is to try to make them more *agentic*. But while agents built from language models are touted as the next big advance [Wang et al., 2024], agentic systems have been identified as a potential source of individual [Dennett, 2023], systemic [Chan et al., 2023, Gabriel et al., 2024], and catastrophic [Ngo et al., 2022] risks. Agency is thus a key focus of behavioural evaluations [Shevlane et al., 2023, Phuong et al., 2024] and governance [Shavit et al., 2023]. Some prominent researchers have even called for a shift towards designing explicitly non-agentic systems [Dennett, 2017, Bengio, 2023].

A critical aspect of agency is the ability to pursue goals. Indeed, the *standard theory of agency* defines agency as the capacity for intentional action – action that can be explained in terms of mental states such as goals [Schlosser, 2019]. But when are we justified in ascribing such mental states? According to Daniel Dennett's instrumentalist philosophy of mind [1989], we are justified when doing so is useful for predicting a system's behaviour.

This paper's key contribution is a method for formally measuring goal-directedness based on Dennett's idea. Since pursuing goals is about having a particular causal effect on the environment[2], defining our measure in a causal model is natural. Causal models are general enough to encompass most frameworks popular among ML practitioners, such as single-decision prediction, classification, and regression tasks, as well as multi-decision (partially observable) Markov decision processes. They also offer enough structure to usefully model many ethics and safety problems [Everitt et al., 2021a, Ward et al., 2024a, Richens et al., 2022, Richens and Everitt, 2024, Everitt et al., 2021b, Ward et al., 2024b, Halpern and Kleiman-Weiner, 2018, Wachter et al., 2017, Kusner et al., 2017, Kenton et al., 2023, Fox et al., 2023, Richens and Everitt, 2024, MacDermott et al., 2023].

---

[1]A notebook that can be used to apply our algorithms in your own tabular MDPs is available at `https://colab.research.google.com/drive/1p_RA7EeN6DWx6x3M4v45R6L84koa4rf_`

[2]This is a common but not universal view in philosophy, see Weirich [2020].

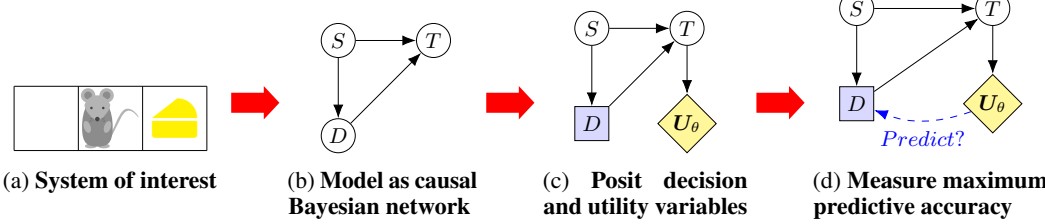

| (a) **System of interest** | (b) **Model as causal Bayesian network** | (c) **Posit decision and utility variables** | (d) **Measure maximum predictive accuracy** |

Figure 1: Computing maximum entropy goal-directedness (MEG).

Maximum entropy goal-directedness (MEG) operationalises goal-directedness as follows, illustrated by the subsequent running example.

> A variable $D$ in a causal model is *goal-directed* with respect to a utility function $\mathcal{U}$ to the extent that the conditional probability distribution of $D$ is well-predicted by the hypothesis that $D$ is optimising $\mathcal{U}$.

**Example 1.** A mouse begins at the centre of a gridworld (Figure 1a). It observes that a block of cheese is located either to the right or left ($S$) with equal probability, proceeds either away from it or towards it ($D$), and thus either obtains the cheese or does not ($T$).

Suppose that the mouse moves left when the cheese is to the left and right when it is to the right, thus consistently obtaining the cheese. Intuitively, this behaviour seems goal-directed, but can we quantify how much? Figure 1 gives an overview of our procedure. We first model the system of interest as a causal Bayesian network (Figure 1b) with variables $S$ for the cheese's position, $D$ for the mouse's movement, and $T$ for whether or not the mouse obtains the cheese. Then, we identify a candidate decision variable $D$ and target variable $T$, hypothesising that the mouse optimises a utility function that depends on $T$ (Figure 1c). Finally, we form a model of what behaviour we should expect from $D$ if it is indeed optimising $U$ and then measure how well this model predicts $D$'s observed behaviour (Figure 1d).

Ziebart [2010]'s maximum causal entropy (MCE) framework suggests a promising way to construct a model for expected behaviour under a given utility function. However, there are several obstacles to applying it to our problem: it cannot measure the predictive usefulness of *known* utility functions, and it only finds the most predictive *linear* utility function. In practice, arbitrary known utility functions can be plugged in, but the results are *not scale-invariant*. We overcome these difficulties by returning to first principles and deriving an updated version of the MCE framework.

In summary, our contributions are four-fold. We (i) adapt the MCE framework to derive *maximum entropy goal-directedness* (MEG), a philosophically-motivated measure of goal-directedness with respect to known utility functions, and show that it satisfies several key desiderata (Section 3); (ii) we extend MEG to measure goal-directedness in cases without a known utility function (Section 4); (iii) we provide algorithms for measuring MEG in MDPs (Section 5); and (iv) we demonstrate the algorithms empirically (Section 6).

**Related Work.** Inverse reinforcement learning (IRL) [Ng and Russell, 2000] focuses on the question of *which* goal a system is optimising, whilst we are interested in *to what extent* the system can be seen as optimising a goal. We are not the first to ask this question. Some works take a Bayesian approach inspired by Dennett's intentional stance.[3] Most closely related to our work is Orseau et al. [2018], which applies Bayesian IRL in POMDPs using a Solomonoff prior over utility functions and an $\varepsilon$-greedy model of behaviour. This lets them infer a posterior probability distribution over whether an observed system is a (goal-directed) "agent" or "just a device". Our approach distinguishes itself from these by considering arbitrary variables in a causal model and deriving our behaviour model from the principle of maximum entropy. Moreover, since our algorithms can take advantage of differentiable classes of utility functions, our approach may be amenable to scaling up using deep neural networks. Oesterheld [2016] combines the intentional stance with Bayes' theorem in cellular automata, but does not consider specific models of behaviour. Like us, Kenton et al. [2023] consider goal-directedness in

---

[3]After accepted publication, another related paper appeared [Xu and Rivera, 2024].

a causal graph. However, they require variables to be manually labelled as *mechanisms* or *object-level*, and only provide a binary distinction between agentic and non-agentic systems (see Appendix A for a detailed comparison). Biehl and Virgo [2022], Virgo et al. [2021] propose a definition of agency in Moore machines based on whether a system's internal state can be interpreted as beliefs about the hidden states of a POMDP.

## 2 Background

We use capital letters for random variables $V$, write $\mathrm{dom}(V)$ for their domain (assumed finite), and use lowercase for outcomes $v \in \mathrm{dom}(V)$. Boldface denotes sets of variables $\boldsymbol{V} = \{V_1, \ldots, V_n\}$, and their outcomes $\boldsymbol{v} \in \mathrm{dom}(\boldsymbol{V}) = \bigtimes_i \mathrm{dom}(V_i)$. Parents and descendants of $V$ in a graph are denoted by $\mathbf{Pa}_V$ and $\mathbf{Desc}_V$, respectively (where $\mathbf{pa}_V$ and $\mathbf{desc}_V$ are their instantiations).

Causal Bayesian networks (CBNs) are a class of probabilistic graphical models used to represent causal relationships between random variables [Pearl, 2009].

**Definition 2.1** (Causal Bayesian network). A *Bayesian network* $M = (G, P)$ over a set of variables $\boldsymbol{V} = \{V_1, \ldots, V_n\}$ consists of a joint probability distribution $P$ which factors according to a directed acyclic graph (DAG) $G$, i.e., $P(V_1, \ldots, V_n) = \prod_{i=1}^{n} P(V_i \mid \mathbf{Pa}_{V_i})$, where $\mathbf{Pa}_{V_i}$ are the parents of $V_i$ in $G$. A Bayesian network is *causal* if its edges represent direct causal relationships, or formally if the result of an intervention $\mathrm{do}(\boldsymbol{X} = \boldsymbol{x})$ for any $\boldsymbol{X} \subseteq \boldsymbol{V}$ can be computed using the *truncated factorisation formula*: $P(\boldsymbol{v} \mid \mathrm{do}(\boldsymbol{X} = \boldsymbol{x})) = \Pi_{i:v_i \notin \boldsymbol{x}} P(v_i \mid \mathbf{pa}_{v_i})$ if $\boldsymbol{v}$ is consistent with $\boldsymbol{x}$ or $P(\boldsymbol{v} \mid \mathrm{do}(\boldsymbol{X} = x)) = 0$ otherwise.

Figure 1b depicts Example 1 as a CBN, showing the causal relationships between the location of the cheese ($S$), the mouse's behavioural response ($D$), and whether the mouse obtains the cheese ($T$).

We are interested in to what extent a set of random variables in a CBN can be seen as goal-directed. That is, to what extent can we interpret them as *decisions* optimising a *utility function*? In other words, we are interested in moving from a CBN to a causal influence diagram (CID), a type of probabilistic graphical model that explicitly identifies decision and utility variables.

**Definition 2.2** (Causal Influence Diagram [Everitt et al., 2021a]). A *causal influence diagram* (CID) $M = (G, P)$ is a CBN where the variables $\boldsymbol{V}$ are partitioned into decision $\boldsymbol{D}$, chance $\boldsymbol{X}$, and utility variables $\boldsymbol{U}$. Instead of a full joint distribution over $\boldsymbol{V}$, $P$ consists of conditional probability distributions (CPDs) for each *non-decision* variable $V \in \boldsymbol{V} \setminus \boldsymbol{D}$.

A CID can be combined with a *policy* $\pi$, which specifies a CPD $\pi_D$ for each decision variable $D$, in order to obtain a full joint distribution. We call the sum of the utility variables the *utility function* and denote it $\mathcal{U} = \sum_{U \in \boldsymbol{U}} U$. Policies are evaluated by their total expected utility $\mathbb{E}_\pi[\mathcal{U}]$. We write $\pi^{\mathrm{unif}}$ for the uniformly random policy.

CIDs can model a broad class of decision problems, including Markov decision processes (MDPs) and partially observable Markov decision processes (POMDPs) [Everitt et al., 2021b].

**Example 2** (POMDP). A mouse begins at the centre of the grid, with a block of cheese located either at the far left or the far right (Figure 2). The mouse does not know its position or the position of the cheese ($S_1$) but can smell which direction the cheese is in ($O_1$) and decide which way to move ($D_1$). Next step, the mouse again smells the direction of the cheese ($O_2$) and again chooses which way to proceed ($D_2$).

## 3 Measuring goal-directedness with respect to a known utility function

**Maximum Entropy Goal-directednes.** Dennett's instrumentalist approach to agency says that we can ascribe mental states (such as utilities) to a system to the extent that doing so is useful for predicting its behaviour [Dennett, 1989]. To operationalise this, we need a model of what behaviour is predicted by a utility function. According to the *principle of maximum entropy* [Jaynes, 1957], we should choose a probability distribution with the highest entropy distribution satisfying our requirements, thus minimising unnecessary assumptions (following Occam's razor). We can measure the entropy of a policy by the expected entropy of its decision variables conditional on their parents $H_\pi(\boldsymbol{D} \parallel \mathbf{Pa}_D) = -\sum_{D \in \boldsymbol{D}} \mathbb{E}_{d, \mathbf{Pa}_D \sim P_\pi} \log \pi_D(d \mid \mathbf{Pa}_D)$. This is Ziebart et al. [2010]'s *causal entropy*, which we usually refer to as just the entropy of $\pi$.

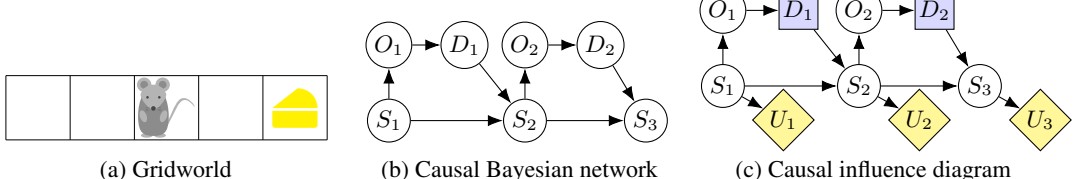

(a) Gridworld      (b) Causal Bayesian network      (c) Causal influence diagram

Figure 2: Sequential multi-decision mouse example.

In our setting, the relevant constraint is expected utility. To avoid assuming that only optimal agents are goal-directed, we construct a set of models of behaviour which covers all levels of competence an agent optimising utility $\mathcal{U}$ could have. We define the set of *attainable expected utilities* in a CID as $\text{att}(\mathcal{U}) = \{u \in \mathbb{R} \mid \exists \pi \in \Pi \left( \mathbb{E}_\pi \left[ \mathcal{U} \right] = u \right) \}$.

**Definition 3.1** (Maximum entropy policy set, known utility function). Let $M = (G, P)$ be a CID with decision variables $\boldsymbol{D}$ and utility function $\mathcal{U}$. The *maximum entropy policy set for $u \in \text{att}(\mathcal{U})$* is $\Pi_{\mathcal{U},u}^{\text{maxent}} := \text{argmax}_{\pi \mid \mathbb{E}_\pi[\mathcal{U}]=u} H_\pi(\boldsymbol{D} \parallel \mathbf{Pa}_D)$. The *maximum entropy policy set for $\mathcal{U}$* is the set of maximum entropy policies for *any* attainable expected utility $\Pi_{\mathcal{U}}^{\text{maxent}} := \bigcup_{u \in \text{att}(\mathcal{U})} \Pi_{\mathcal{U},u}^{\text{maxent}}$.

For each attainable expected utility, $\Pi_{\mathcal{U}}^{\text{maxent}}$ contains the highest entropy policy which attains it. In MDPs, this policy is unique $\pi_{\mathcal{U},u}^{\text{maxent}}$ and can be found with backwards induction (see Section 5).

We measure predictive accuracy using cross-entropy, as is common in ML. We subtract the predictive accuracy of the uniform distribution, so that we measure predictive accuracy relative to random chance. This makes MEG always non-negative.

**Definition 3.2** (Maximum entropy goal-directedness, known utility function). Let $M := (G, P)$ be a CID with decision variables $\boldsymbol{D}$ and utility function $\mathcal{U}$. The *maximum entropy goal-directedness* (MEG) of a policy $\pi$ with respect to $\mathcal{U}$ is

$$\text{MEG}_{\mathcal{U}}(\pi) := \max_{\pi^{\text{maxent}} \in \Pi_{\mathcal{U}}^{\text{maxent}}} \mathbb{E}_\pi \left[ \sum_{D \in \boldsymbol{D}} \left( \log \pi^{\text{maxent}}(D \mid \mathbf{Pa}_D) - \log \frac{1}{|\text{dom}(D)|} \right) \right]. \quad (1)$$

As we will see, in a Markov Decision Process, this is equivalent to maximising predictive accuracy by varying the rationality parameter $\beta$ in the 'soft-optimal' policy $\pi_\beta^{\text{soft}}$ used in maximum entropy reinforcement learning [Haarnoja et al., 2017].[4]

If instead of having access to a policy $\pi$, we have access to a set of full trajectories $\{(\mathbf{pa}_{D_1}^i, D_1^i, \ldots, \mathbf{pa}_{D_n^i}, D_n^i)\}_i$, the expectation $\mathbb{E}_\pi$ in Equation (1) can be replaced with an average over the trajectory set. This is an unbiased and consistent estimate of $\text{MEG}_{\mathcal{U}}(\pi)$ for the policy $\pi$ generating the trajectories.

**Example.** Consider a policy $\pi$ in Example 1 that proceeds towards the cheese with probability 0.8. How goal-directed is this policy with respect to the utility function $\mathcal{U}$ that gives $+1$ for obtaining the cheese and $-1$ otherwise?

To compute $\text{MEG}_{\mathcal{U}}(\pi)$, we first find the maximum entropy policy set $\Pi_{\mathcal{U}}^{\text{maxent}}$, and then take the maximum predictive accuracy with respect to $\pi$. In a single-decision setting, for each attainable expected utility $u$ there is a unique $\pi_{\mathcal{U},u}^{\text{maxent}}$ which has the form of a Boltzmann policy $\pi_{\mathcal{U},u}^{\text{maxent}}(d \mid s) = \frac{\exp(\beta \cdot \mathbb{E}[U|d,s])}{\sum_{d'} \exp(\beta \cdot \mathbb{E}[U|d',s])}$ (cf. Theorem 5.1). The rationality parameter $\beta = \beta(u)$ can be varied to get the correct expected utility. Predictive accuracy with respect to $\pi$ is maximised by $\pi_{\mathcal{U},0.8}^{\text{maxent}}$, which has a rationality parameter of $\beta = \log 2$. The expected logprob of a prediction of this policy is

---

[4]A noteworthy property of Definition 3.2 is that a policy that *pessimises* a utility function can be just as goal-directed as a policy that *optimises* it. This fits with the intuitive notion that we are measuring how likely a policy's performance on a particular utility function is to be an accident. However, it might be argued that a policy should instead be considered *negatively* goal-directed with respect to a utility function it pessimises. We could adjust Definition 3.2 to achieve this by multiplying by the sign of $\mathbb{E}_\pi \left[ \mathcal{U} \right] - \mathbb{E}_{\pi^{\text{unif}}} \left[ \mathcal{U} \right]$, making the goal-directedness of any policy that performs worse than random negative. For simplicity, we use the unsigned measure in this paper.

$\mathbb{E}_\pi \left[ \log \pi_{\mathcal{U},0.8}^{\mathrm{maxent}}(D \mid \mathbf{Pa}_D) \right] = -0.50$, while the expected logprob of a uniform prediction is $\log(\frac{1}{2}) = -0.69$. So we get that $\mathrm{MEG}_\mathcal{U}(\pi) = -0.50 - (-0.69) = 0.19$. For comparison, predictive accuracy for the optimal policy $\pi^*$ is maximised when $\beta = \infty$, and has $\mathrm{MEG}_\mathcal{U}(\pi^*) = 0 - (-0.69) = 0.69$.

**Properties.** We now show that MEG satisfies three important desiderata. First, since utility functions are usually only defined up to translation and rescaling, a measure of goal-directedness with respect to a utility function should be translation and scale invariant. MEG satisfies this property:

**Proposition 3.1** (Translation and scale invariance). *Let $M_1$ be a CID with utility function $\mathcal{U}_1$, and let $M_2$ be an identical CID but with utility function $\mathcal{U}_2 = a \cdot \mathcal{U}_1 + b$, for some $a, b \in \mathbb{R}$, with $a \neq 0$. Then for any policy $\pi$, $\mathrm{MEG}_{\mathcal{U}_1}(\pi) = \mathrm{MEG}_{\mathcal{U}_2}(\pi)$.*

Second, goal-directedness should be minimal when actions are chosen completely at random and maximal when uniquely optimal actions are chosen.

**Proposition 3.2** (Bounds). *Let $M$ be a CID with utility function $\mathcal{U}$. Then for any policy $\pi$ we have $0 \leq \mathrm{MEG}_\mathcal{U}(\pi) \leq \sum_{D \in \mathcal{D}} \log(|\mathrm{dom}(D)|)$, with equality in the lower bound if $\pi$ is the uniform policy, and equality in the upper bound if $\pi$ is the unique optimal (or anti-optimal) policy with respect to $\mathcal{U}$.*

Note that MEG has a natural interpretation as the amount of evidence provided for a goal-directed policy over a purely random policy. The larger a decision problem, the more opportunity there is to see this evidence, so the higher MEG can be.

Third, a system can never be goal-directed towards a utility function it cannot affect.

**Proposition 3.3** (No goal-directedness without causal influence). *Let $M = (G, P)$ be a CID with utility function $\mathcal{U}$ and decision variables $\mathbf{D}$ such that, $\mathbf{Desc}(\mathbf{D}) \cap \mathbf{Pa}_U = \emptyset$. Then $\mathrm{MEG}_\mathcal{U}(\mathbf{D}) = 0$.*

**Comparison to MCE IRL** Our method is closely related to MCE IRL [Ziebart et al., 2010] (see Gleave and Toyer [2022] for a useful primer). In this subsection, we discuss the key similarities and differences.

The MCE IRL method seeks to find a utility function that explains the policy $\pi$. It starts by identifying a set of $n$ linear features $f_i$ and seeks a model policy that imitates $\pi$ as far as these features are concerned but otherwise is as random as possible. It thus applies the principle of maximum entropy with $n$ linear constraints. The form of the model policy involves a weighted sum of these features. In a single-decision example, it takes the form

$$\pi^{\mathrm{MCE}}(d \mid s) = \frac{\exp\left(\mathbb{E}\left[\sum_i w_i f_i \mid d, s\right]\right)}{\sum_{d'} \exp\left(\mathbb{E}\left[w_i f_i \mid d', s\right]\right)}. \tag{2}$$

The weights $w_i$ are interpreted as a utility function over the features $f_i$. MCE IRL can, therefore, only return a linear utility function.

In contrast, our method seeks to measure the goal-directedness of $\pi$ with respect to an arbitrary utility function $\mathcal{U}$, linear or otherwise. Rather than constructing a single maximum entropy policy with $n$ linear constraints, we construct a class of maximum entropy policies, each with a different single constraint on the expected utility.

A naive alternative to defining the goal-directedness of $\pi$ with respect to $\mathcal{U}$ as the maximum predictive accuracy across $\mathcal{U}$'s maximum policy set, we could simply plug in our utility function $\mathcal{U}$ to $\pi^{\mathrm{MCE}}$ from Equation (2), and use that to measure predictive accuracy. If $\mathcal{U}$ is linear in the features $f_i$, we could substitute in the appropriate weights, but even if not, we could still replace $\sum_i w_i f_i$ with $\mathcal{U}$. Indeed, this is often done with nonlinear utility functions in *deep* MCE IRL [Wulfmeier et al., 2015].

However, this would not have a formal justification, and we would run into a problem: scale non-invariance. Plugging in $2 \cdot \mathcal{U}$ would result in a more sharply peaked $\pi^{\mathrm{MCE}}$ than $\mathcal{U}$; in Example 1, we would get that the mouse is more goal-directed towards $2 \cdot \mathcal{U}$ than $\mathcal{U}$, with a predictive accuracy (measured by negative cross-entropy) of -0.018 vs -0.13. In contrast, constructing separate maximum entropy policies for each expected utility automatically handles this issue. The policy in $\Pi_{2 \cdot \mathcal{U}}^{\mathrm{maxent}}$ which maximises predictive accuracy for $\pi$ has an inversely scaled rationality parameter $\beta' = \frac{\beta}{2}$ compared to the maximally predictive policy in $\Pi_{2 \cdot \mathcal{U}}^{\mathrm{maxent}}$. In other words, they are the same policy, and we get that $\mathrm{MEG}_\mathcal{U}(\pi) = \mathrm{MEG}_{2 \cdot \mathcal{U}}(\pi) = 0.19$ (cf. Proposition 3.1).

# 4   Measuring goal-directedness without a known utility function

In many cases where we want to apply MEG, we may not know exactly what utility function the system could be optimising. For example, we might suspect that a content recommender is trying to influence a user's preferences, but may not have a clear hypothesis as to in what way. Therefore, in this section, we extend our definitions for measuring goal-directedness to the case where the utility function is unknown.

We first extend CIDs to include multiple possible utility functions.

**Definition 4.1.** A *parametric-utility CID* (CID) $M^\Theta$ is a set of CIDs $\{M^\theta \mid \theta \in \Theta\}$ which differ only in the CPDs of their utility variables.

In effect, a parametric CID is a CID with a parametric class of utility functions $\mathcal{U}^\Theta$.

The maximum entropy policy set from Definition 3.1 is extended accordingly to include maximum entropy policies *for each utility function* and each attainable expected utility with respect to it.

**Definition 4.2** (Maximum entropy policy set, unknown utility function). Let $M^\Theta = (G, P)$ be a parametric-utility CID with decision variables $\boldsymbol{D}$ and utility function $\mathcal{U}^\Theta$. The *maximum entropy policy set for $\mathcal{U}^\Theta$* is the set of maximum entropy policies for any attainable expected utility for any utility function in the class: $\Pi^{\mathrm{maxent}}_{\mathcal{U}^\Theta} := \bigcup_{\theta \in \Theta, u \in \mathrm{att}(\mathcal{U}^\theta)} \Pi^{\mathrm{maxent}}_{\mathcal{U}^\theta, u}$.

**Definition 4.3** (MEG, unknown utility function). Let $M^\Theta = (G, P)$ be a parametric-utility CID with decision variables $\boldsymbol{D}$ and utility function $\mathcal{U}^\Theta$. The *maximum entropy goal-directedness* of a policy $\pi$ with respect to $\mathcal{U}^\Theta$ is $\mathrm{MEG}_{\mathcal{U}^\Theta}(\pi) = \max_{\mathcal{U} \in \mathcal{U}^\Theta} \mathrm{MEG}_{\mathcal{U}}(\pi)$.

**Definition 4.4** (MEG, target variables). Let $M = (G, P)$ be a CBN with variables $\boldsymbol{V}$. Let $\boldsymbol{D} \subseteq \boldsymbol{V}$ be a hypothesised set of decision variables and $\boldsymbol{T} \subseteq V$ be a hypothesised set of *target* variables. The *maximum entropy goal-directedness* of $\boldsymbol{D}$ with respect to $\boldsymbol{T}$, $\mathrm{MEG}_{\boldsymbol{T}}(\boldsymbol{D})$, is the goal-directedness of $\pi = P(\boldsymbol{D} \mid \mathbf{Pa}_D)$ in the parametric CID with decisions $\boldsymbol{D}$ and utility functions $\mathcal{U} : \mathrm{dom}(\boldsymbol{T}) \to \mathbb{R}$ (i.e., the set of all utility functions over $T$).

For example, if we only suspected that the mouse in Example 1 was optimising some function of the cheese $T$, but didn't know which one, we could apply Definition 4.4 to consider the goal-directedness towards $T$ under any utility function defined on $T$. Thanks to translation and scale invariance (Proposition 3.1), there are effectively only three utility functions to consider: those that provide higher utility to cheese than not cheese, those that do the opposite, and those that are indifferent.

Note that $\boldsymbol{T}$ has to include some descendants of $\boldsymbol{D}$, in order to enable positive MEG (Proposition 3.3). However, it is not necessary that $\boldsymbol{T}$ consists of *only* descendants of $\boldsymbol{D}$ (i.e., $\boldsymbol{T}$ need not be a subset of $\mathbf{Desc}(\boldsymbol{D})$). For example, goal-conditional agents take an instruction as part of their input $\mathbf{Pa}_D$. The goal-directedness of such agents can only be fully appreciated by including the instruction in $\boldsymbol{T}$.

**Pseudo-terminal goals.**   Definition 4.4 enable us to state a result about a special kind of instrumental goal. It is well known that an agent that optimises some variable has an instrumental incentive to control any variables which mediate between the two [Everitt et al., 2021a]. However, since the agent might want the mediators to take different values in different circumstances, it need not appear goal-directed with respect to the mediators. Theorem 4.1 shows that in the special case where the mediators *d-separate* the decision from the downstream variable, the decision appears at least as goal-directed with respect to the mediators as with respect to the target.

**Theorem 4.1** (Pseudo-terminal goals). *Let $M = ((\boldsymbol{V}, \boldsymbol{E}), P)$ be a CBN. Let $\boldsymbol{D}, \boldsymbol{T}, \boldsymbol{S} \subseteq \boldsymbol{V}$ such that $\boldsymbol{D} \perp \boldsymbol{T} \mid \boldsymbol{S}$. Then $\mathrm{MEG}_{\boldsymbol{T}}(\boldsymbol{D}) \leq \mathrm{MEG}_{\boldsymbol{S}}(\boldsymbol{D})$.*

For example, in Figure 2, the agent must be at least as goal-directed towards $S_3$ as it is towards $U_3$, since $S_3$ blocks all paths (i.e. d-separates) from $\{D_1, D_2\}$ to $U_3$.

Intuitively, this comes about because, in such cases, a rational agent wants the mediators to take the same values regardless of circumstances, making the instrumental control incentive indistinguishable from a terminal goal. This means we do not have to look arbitrarily far into the future to find evidence of goal-directedness. An agent that is goal-directed with respect to next week must also be goal-directed with respect to tomorrow.

# 5 Computing MEG in Markov Decision Processes

In this section, we give algorithms for computing MEG in MDPs. First, we define what an MDP looks like as a CID. We then establish that soft value iteration can be used to construct our maximum entropy policies, and give algorithms for computing MEG when the utility function is known or unknown.

Note that in order to run these algorithms, we do not need an explicit causal model, and so we do not have to worry about hidden confounders. We do, however, need black-box access to the environment dynamics, i.e., the ability to run different policies in the environment and measure whatever variables we are considering utility functions over.

**Definition 5.1.** A *Markov Decision Process* (MDP) is a CID with variables $\{S_t, D_t, U_t\}_{t=1}^n$, decisions $\boldsymbol{D} = \{D_t\}_{t=1}^n$ and utilities $\boldsymbol{U} = \{U_t\}_{t=1}^n$, such that for $t$ between 1 and $n$, $\mathbf{Pa}_{D_t} = \{S_t\}$, $\mathbf{Pa}_{U_t} = \{S_t\}$, while $\mathbf{Pa}_{S_t} = \{S_{t-1}, D_{t-1}\}$ for $t > 1$, and $\mathbf{Pa}_{S_1} = \emptyset$.

**Constructing Maximum Entropy Policies**   In MDPs, Ziebart's soft value iteration algorithm can be used to construct maximum entropy policies satisfying a set of linear constraints. We apply it to construct maximum entropy policies satisfying expected utility constraints.

**Definition 5.2** (Soft Q-Function).   Let $M = (G, P)$ be an MDP. Let $\beta \in \mathbb{R} \setminus \{0\}$. For each $D_t \in \boldsymbol{D}$ we define the *soft Q-function* $Q_{\beta,n}^{\text{soft}} : \text{dom}(D_t) \times \text{dom}(\mathbf{Pa}_{D_t}) \to \mathbb{R}$ via the recursion:

$$Q_{\beta,t}^{\text{soft}}(d_t \mid \mathbf{pa}_t) = \mathbb{E}\left[ U_t + \frac{1}{\beta} \text{logsumexp}(\beta \cdot Q_{\beta,t+1}^{\text{soft}}(\cdot \mid \mathbf{Pa}_{D_{t+1}})) \Big| d_t, \mathbf{pa}_{t+1} \right] \quad \text{for } t < n,$$

$$Q_{\beta,n}^{\text{soft}}(d_n \mid \mathbf{pa}_n) = \mathbb{E}\left[ U_n \mid d_n, \mathbf{pa}_n \right],$$

where $\text{logsumexp}\,\beta(Q_{\beta,t+1}^{\text{soft}}(\cdot \mid \mathbf{Pa}_{D_{t+1}})) = \log \sum_{d_{t+1} \in \text{dom}(D_{t+1})} \exp(\beta Q_{\beta,t+1}^{\text{soft}}(d_{t+1} \mid \mathbf{Pa}_{D_{t+1}}))$.

Using the soft Q-function, we show that there is a unique $\pi \in \Pi_{\mathcal{U},u}^{\text{maxent}}$ for each $\mathcal{U}$ and $u$ in MDPs.

**Theorem 5.1** (Maximum entropy policy in MDPs). *Let $M = (G, P)$ be an MDP with utility function $\mathcal{U}$, and let $u \in \text{att}(\mathcal{U})$ be an attainable expected utility. Then there exists a unique maximum entropy policy $\pi_u^{\text{maxent}} \in \Pi_{\mathcal{U},u}^{\text{maxent}}$, and it has the form*

$$\pi_{u,t}^{\text{maxent}}(d_t \mid \boldsymbol{pa}_t) = \pi_{\beta,t}^{\text{soft}}(d_t \mid \boldsymbol{pa}_t) := \begin{cases} \frac{\exp(\beta \cdot Q_{\beta,t}^{\text{soft}}(d_t \mid \boldsymbol{pa}_t))}{\sum_{d' \in \text{dom}(D_t)} \exp(\beta \cdot Q_{\beta,t}^{\text{soft}}(d'_t \mid \boldsymbol{pa}_t))}, & \text{if } \beta \neq 0 \\ \pi^{\text{unif}}(d_t \mid \boldsymbol{pa}_t), & \text{if } \beta = 0. \end{cases} \quad (3)$$

*where $\beta = \text{argmax}_{\beta' \in \mathbb{R} \cup \{\infty, -\infty\}} \sum_t \mathbb{E}_\pi \left[ \log(\pi_{\beta',t}^{\text{soft}}(d_t \mid \boldsymbol{pa}_t)) \right]$.*

We refer to a policy of the form of $\pi_\beta$ as a soft-optimal policy with a rationality parameter of $\beta$.

**Known Utility Function**   To apply Definition 3.1 to measure the goal-directedness of a policy $\pi$ in a CID $M$ with respect to a utility function $\mathcal{U}$, we need to find the maximum entropy policy in $\Pi_{\mathcal{U}}^{\text{maxent}}$ which best predicts $\pi$. We can use Theorem 5.1 to derive an algorithm that finds $\pi_u^{\text{maxent}}$ for any $u \in \text{att}(\mathcal{U})$.

Fortunately, we do not need to consider each policy in $\Pi_{\mathcal{U},u}^{\text{maxent}}$ individually. We know the form of $\pi_u^{\text{maxent}}$, and only the real-valued rationality parameter $\beta$ varies depending on $u$.

The gradient of the predictive accuracy with respect to $\beta$ is

$$\nabla_\beta \mathbb{E}_\pi \left[ \sum_{D \in \boldsymbol{D}} \left( \log \pi_\beta^{\text{soft}}(D \mid \boldsymbol{Pa_D}) - \log \frac{1}{|\text{dom}(D)|} \right) \right] = \mathbb{E}_\pi [\mathcal{U}] - \mathbb{E}_{\pi_\beta^{\text{soft}}} [\mathcal{U}].$$

The predictive accuracy is a concave function of $\beta$, so we can apply gradient ascent to find the global maximum in $\beta$, which is the same as finding the maximum in $u$.

$\text{MEG}_{\mathcal{U}}(\pi)$ can therefore be found by alternating between applying the soft value iteration of Definition 5.2 to find $\pi_\beta^{\text{maxent}}$, computing $\mathbb{E}_\pi [\mathcal{U}] - \mathbb{E}_{\pi_\beta^{\text{maxent}}} [\mathcal{U}]$, and taking a gradient step. See Algorithm 1.

If the $\beta$ that maximises predictive accuracy is $\infty$ or $-\infty$, which can happen if $\pi$ is optimal or anti-optimal with respect to $\mathcal{U}$, then the algorithm can never reach the (nonetheless finite) value of $\text{MEG}_{\mathcal{U}}(\pi)$, but will still converge in the limit.

---

**Algorithm 1** Known-utility MEG in MDPs

---

**Input:** MDP $M$, policy $\pi$
**Output:** $\text{MEG}_{\mathcal{U}}(\pi)$
1: initialise rationality parameter $\beta$, set learning rate $\alpha$.
2: **repeat**
3:     Apply soft value iteration to find $Q_\beta^{\text{soft}}$            ▷ Definition 5.2
4:     $\pi_\beta^{\text{soft}} \leftarrow \text{softmax}(\beta \cdot Q_\beta^{\text{soft}})$
5:     $g \leftarrow \left( \mathbb{E}_\pi \left[ \mathcal{U} \right] - \mathbb{E}_{\pi_\beta^{\text{soft}}} \left[ \mathcal{U} \right] \right)$
6:     $\beta \leftarrow \beta + \alpha \cdot g$
7: **until** Convergence
8: **return** $\mathbb{E}_\pi \left[ \sum_{D \in \boldsymbol{D}} \left( \log \pi_\beta^{\text{soft}}(D \mid \mathbf{Pa}_D) - \log \frac{1}{|\text{dom}(D)|} \right) \right]$

---

**Unknown-utility algorithm** To find unknown-utility MEG, we consider soft-optimal policies for various utility functions $\mathcal{U}$ and maximise predictive accuracy with respect to $\mathcal{U}$ as well as $\beta$. Let $\mathcal{U}^\Theta$ be a differentiable class of functions, such as a neural network, and write $\pi_{\theta,\beta}^{\text{soft}}$ for a soft-optimal policy with respect to $\mathcal{U} \in \mathcal{U}^\Theta$, and $Q_{\theta,\beta}^{\text{soft}}$ for the corresponding soft Q-function. We can take the derivative of the predictive accuracy with respect to $\theta$ and get $\mathbb{E}_\pi \left[ \nabla_\theta \mathcal{U} \right] - \mathbb{E}_{\pi_\beta^{\text{maxent}}} \left[ \nabla_\theta \mathcal{U} \right]$. Algorithm 2 extends Algorithm 1 to this case.

---

**Algorithm 2** Unknown-utility MEG in MDPs

---

**Input:** Parametric MDP $M_\Theta$ over differentiable class of utility functions, policy $\pi$
**Output:** $\text{MEG}_{\mathcal{U}_\Theta}(\pi)$
1: Initialise utility parameter $\theta$, rationality parameter $\beta$, set learning rate $\alpha$.
2: **repeat**
3:     Apply soft value iteration to find $Q_{\theta,\beta}^{\text{soft}}$            ▷ Definition 5.2
4:     $\pi_{\theta,\beta}^{\text{soft}} \leftarrow \text{softmax}(\beta \cdot Q_{\theta,\beta}^{\text{soft}})$
5:     $g_\beta \leftarrow \left( \mathbb{E}_\pi \left[ \mathcal{U}^\theta \right] - \mathbb{E}_{\pi_\beta^{\text{soft}}} \left[ \mathcal{U}^\theta \right] \right)$
6:     $g_\theta \leftarrow \left( \mathbb{E}_\pi \left[ \nabla_\theta \mathcal{U}^\theta \right] - \mathbb{E}_{\pi_\beta^{\text{soft}}} \left[ \nabla_\theta \mathcal{U}^\theta \right] \right)$
7:     $\beta \leftarrow \beta + \alpha \cdot g_\beta$
8:     $\theta \leftarrow \beta + \alpha \cdot g_\theta$
9: **until** Stopping condition
10: **return** $\mathbb{E}_\pi \left[ \sum_{D \in \boldsymbol{D}} \left( \log \pi_{\theta,\beta}^{\text{soft}}(D \mid \mathbf{Pa}_D) - \log \frac{1}{|\text{dom}(D)|} \right) \right]$

---

An important caveat is that if $\mathcal{U}^\theta$ is a non-convex function of $\theta$ (e.g. a neural network), Algorithm 2 need not converge to a global maximum. In general, the algorithm provides a *lower bound* for $\text{MEG}_{\mathcal{U}_\theta}(\pi)$, and hence for $\text{MEG}_{\boldsymbol{T}}(\pi)$ where $\boldsymbol{T} = \mathbf{Pa}_{\mathcal{U}^\Theta}$. In practice, we may want to estimate the soft Q-function and expected utilities with Monte Carlo or variational methods, in which case the algorithm provides an *approximate* lower bound on goal-directedness.

## 6 Experimental Evaluation

We carried out two experiments[5] to measure known-utility MEG with respect to the environment reward function and unknown-utility MEG with respect to a hypothesis class of utility functions. We used an MLP with a single hidden layer of size 256 to define a utility function over states.

Our experiments measured MEG for various policies in the CliffWorld environment from the seals suite [Gleave et al., 2020]. Cliffworld (Figure 3a) is a 4x10 gridworld MDP in which the agent starts in the top left corner and aims to reach the top right while avoiding the cliff along the top row. With probability 0.3, wind causes the agent to move upwards by one more square than intended. The

---

[5]Code available at `https://github.com/mattmacdermott1/measuring-goal-directedness`

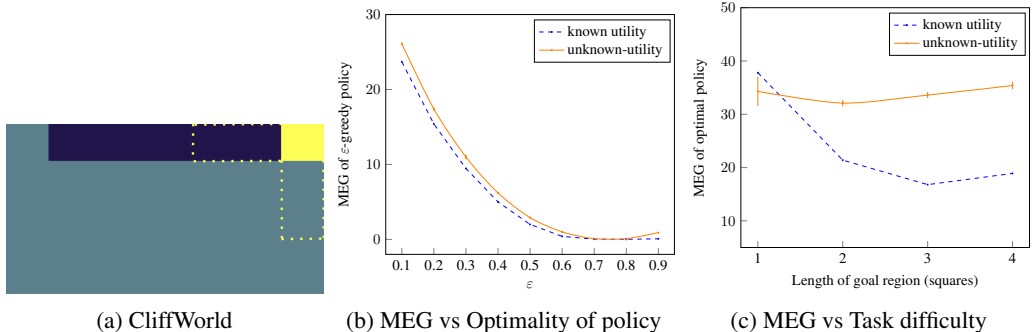

|              (a) CliffWorld              |       (b) MEG vs Optimality of policy       |        (c) MEG vs Task difficulty        |

Figure 3: (a) The CliffWorld environment. (b) MEG of $\varepsilon$-greedy policies for varying $\varepsilon$. MEG decreases as the policy gets less optimal. (c) MEG for optimal policies for various reward functions. Known-utility MEG decreases as the goal gets easier to satisfy, but unknown-utility MEG stays higher because the optimal policies also do well with respect to a narrower goal.

environment reward function gives $+10$ when the agent is in the (yellow) goal square, $-10$ for the (dark blue) cliff squares, and $-1$ elsewhere. The dotted yellow line indicates a length 3 goal region.

**MEG vs Optimality of policy**  In our first experiment, we measured the goal-directedness of policies of varying degrees of optimality by considering $\varepsilon$-greedy policies for $\varepsilon$ in the range $0.1$ to $0.9$. Figure 3b shows known- and unknown utility MEG for each policy.[6] Predictably, the goal-directedness with respect to the environment reward decreased toward $0$ as the policy became less optimal. So did unknown-utility MEG – since, as $\varepsilon$ increases, the policy becomes increasingly uniform, it does not appear goal-directed with respect to *any* utility function over states.

**MEG vs Task difficulty**  In our second experiment, we measured the goal-directedness of optimal policies for reward functions of varying difficulty. We extended the goal region of Cliffworld to run for either 1, 2, 3 or 4 squares along the top row and back column and considered optimal policies for each reward function. Figure 3c shows Cliffworld with a goal region of length 3. Figure 3b shows the results. Goal-directedness with respect to the true reward function decreased as the task became easier to complete. A way to interpret this is that as the number of policies which do well on a reward function increases, doing well on that reward function provides less and less evidence for deliberate optimisation. In contrast, unknown-utility MEG stays high even as the environment reward becomes easier to satisfy. This is because the optimal policy proceeds towards the *nearest* goal-squares and, hence, it appears strongly goal-directed with respect to a utility function which gives high reward to only those squares. Since this narrower utility function is more difficult to do well on than the environment reward function, doing well on it provides more evidence for goal-directedness. In Appendix D.3, we visualise the policies in question to make this more explicit. We also give tables of results for both experiments.

## 7 Limitations

**Environment Access**  Although computing MEG does not require an explicit causal model (cf. Section 5), it does require the ability to run various policies in the environment of interest – we cannot compute MEG purely from observational data.

**Intractability**  While MEG can be computed with gradient descent, doing so may well be computationally intractable in high dimensional settings. In this paper, we conduct only preliminary experiments – larger experiments based on real-world data may explore how serious these limitations are in practice.

**Choice of variables**  MEG is highly dependent on which variables we choose to measure goal-directedness with respect to. At one extreme, all (deterministic) policies have maximal goal-

---

[6]Known-utility MEG is deterministic. Unknown-utility MEG depends on the random initialisation of the neural network, so we show the mean of several runs. Full details are given in Appendix D.3.

directedness with respect to their own actions (given an expressive enough class of utility functions). This means that, for example, it would not be very insightful to compute the goal-directedness of a language model with respect to the text it outputs [7]. At the other extreme, if a policy is highly goal-directed towards some variable $T$ that we do not think to measure, MEG may be misleadingly low. Relatedly, MEG may also be affected by whether we use a binary variable for $T$ (or the decisions $D$) rather than a fine-grained one with many possible outcomes. We should, therefore, think of MEG as measuring what *evidence* a set of variables provides about a policy's goal-directedness. A lack of evidence does not necessarily mean a lack of goal-directedness.

**Distributional shifts**   MEG measures how predictive a utility function is of a system's behaviour *on distribution*, and distributional shifts can lead to changes in MEG. A consequence of this is that two almost identical policies can be given arbitrarily different goal-directedness scores, for example, if they take different actions in the start state of an MDP and thus spend time in different regions of the state space. It may be that by considering changes to a system's behaviour under interventions, as Kenton et al. [2023] do, we can distinguish "true" goals from spurious goals, where the former predict behaviour well across distributional shifts, while the latter happen to predict behaviour well on a particular distribution (perhaps because they correlate with true goals) [Di Langosco et al., 2022]. We leave this to future work.

**Behavioural vs mechanistic approaches**   The latter two points suggest that mechanistic approaches to understanding a system's goal-directedness could have advantages over behavioural approaches like MEG. For example, suppose we could reverse engineer the algorithms learned by a neural network. In that case, it may be possible to infer which variables the system is goal-directed with respect to and do so in a way which is robust to distributional shifts. However, mechanistic interpretability of neural networks [Elhage et al., 2021] is an ambitious research agenda that is still in its infancy, and for now, behavioural approaches appear more tractable.

**Societal implications**   An empirical measure of goal-directedness could enable researchers and companies to keep better track of how goal-directed LLMs and other systems are. This is important for dangerous capability evaluations [Shevlane et al., 2023, Phuong et al., 2024] and governance [Shavit et al., 2023]. A potential downside is that it could enable bad actors to create more dangerous systems by optimising for goal-directedness. We judge that this does not contribute much additional risk, given that bad actors can already optimise a system to pursue actively harmful goals.

# 8   Conclusion

We proposed maximum entropy goal-directedness (MEG), a formal measure of goal-directedness in CIDs and CBNs, grounded in the philosophical literature and the maximum causal entropy principle. Developing such measures is important because many risks associated with advanced artificial intelligence come from goal-directed behaviour. We proved that MEG satisfies several key desiderata, including scale invariance and being zero with respect to variables that can't be influenced, and that it gives insights about instrumental goals. On the practical side, we adapted algorithms from the maximum causal entropy framework for inverse reinforcement learning to measure goal-directedness with respect to nonlinear utility functions in MDPs. The algorithms handle both a single utility function and a differentiable class of utility functions. The algorithms were used in some small-scale experiments measuring the goal-directedness of various policies in MDPs. In future work, we plan to develop MEG's practical applications further. In particular, we hope to apply MEG to neural network interpretability by measuring the goal-directedness of a neural network agent with respect to a hypothesis class of utility functions taking the network's hidden states as input, thus taking more than just the system's behaviour into account.

---

[7]Any proposed way of measuring goal-directedness needs a way to avoid the trivial result where all policies are seen to be maximising an idiosyncratic utility function that 'overfits' to that policy's behaviour. MEG avoids this by allowing us to measure goal-directedness with respect to a set of variables that excludes the system's actions (for example, states in an MDP). An alternative approach is to penalise more complex utility functions.

# 9 Acknowledgements

The authors would like to thank Ryan Carey, David Hyland, Laurent Orseau, Francis Rhys Ward, and several anonymous reviewers for useful feedback. This research is supported by the UKRI Centre for Doctoral Training in Safe and Trusted AI (EP/S023356/1), and by the EPSRC grant "An Abstraction-based Technique for Safe Reinforcement Learning" (EP/X015823/1). Fox acknowledges the support of the EPSRC Centre for Doctoral Training in Autonomous Intelligent Machines and Systems (EP/S024050/1).

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

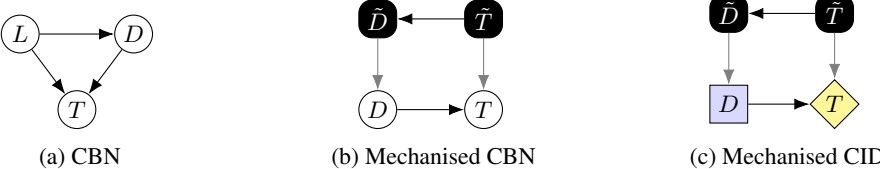

|  (a) CBN | (b) Mechanised CBN | (c) Mechanised CID |

Figure 4: Example 1 can be equally well represented with a CBN (a) or mechanised CBN (b), but Kenton et al. [2023]'s algorithm only identifies an agent in (b). (c) shows the resulting mechanised CID. In contrast, MEG is unchanged between (b) and (c). Note also that the causal discovery algorithm identifies $T$ as a utility variable, where where MEG adds a new utility child to $T$.

## A  Comparison to Discovering Agents

This paper is inspired by Kenton et al. [2023], who proposed a causal discovery algorithm for identifying agents in causal models, inspired by Daniel Dennett's view of agents as systems "moved by reasons" [Dennett, 1989]. Our approach has several advantages over theirs, which we enumerate below.

**Mechanism variables.** Kenton et al. [2023] assume access to a *mechanised structural causal model*, which augments an ordinary causal model with *mechanism variables* which parameterise distributions of ordinary object-level variables. An agent is defined as a system that adapts to changes in the mechanism of its environment. However, the question of what makes a variable a mechanism is left undefined, and indeed, the same causal model can be expressed either with or without mechanism variables, leading their algorithm to give a different answer. For example, Example 1 has identical causal structure to the mouse example in Kenton et al. [2023], but without any variables designated as mechanisms. Their algorithm says the version with mechanism variables contains an agent while the version without does not, despite them being essentially the same causal model. Figure 4 shows our example depicted as a mechanised structural causal model. We fix this arbitrariness by making our definition in ordinary causal Bayesian networks.

**Utility variables.** Their algorithm assumes that some variables in the model represent agents' utilities. We bring this more in line with the philosophical motivation by treating utilities as hypothesised mental states with which we augment our model.

**Predictive accuracy.** Kenton et al. [2023]'s approach formalises Dennett's idea of agents as systems "moved by reasons". We build on this idea but bring it more in line with Dennett's notion of what it means for a system to be moved by a reason – that the reason is useful for predicting its behaviour.

**Gradualist vs Essentialist.** The predictive error viewpoint gives us a continuous measure of goal-directedness rather than a binary notion of agency, which is more befitting of the gradualist view of agents which inspired it.

**Practicality.** Their algorithm is theoretical rather than something that can be applied in practice. Instead, ours is straightforward to implement, as we demonstrate in Section 6. This opens up a range of potential applications, including behavioural evaluations and interpretability of ML models.

**Interventional distributions.** The primary drawback of MEG is that it doesn't necessarily generalise outside of the distribution. Running MEG on interventional distributions may fix this. We leave an extension of MEG to interventional distributions for future work.

## B  Proofs of MEG Properties

**Proposition 3.1** (Translation and scale invariance). *Let $M_1$ be a CID with utility function $\mathcal{U}_1$, and let $M_2$ be an identical CID but with utility function $\mathcal{U}_2 = a \cdot \mathcal{U}_1 + b$, for some $a, b \in \mathbb{R}$, with $a \neq 0$. Then for any policy $\pi$, $\mathrm{MEG}_{\mathcal{U}_1}(\pi) = \mathrm{MEG}_{\mathcal{U}_2}(\pi)$.*

*Proof.* Since MEG is defined by maximising over a maximum entropy policy set, showing that two utility functions have the same maximum entropy policy set is enough to show that they result in the same MEG for every policy.

If $\pi \in \Pi^{\text{maxent}}_{\mathcal{U}_1}$, then $\pi \in \Pi^{\text{maxent}}_{\mathcal{U}_1, u_1}$ for some $u_1 \in \text{att}(\mathcal{U}_1)$, i.e. $\pi$ is a maximum entropy policy satisfying the constraint that $\mathbb{E}_\pi[\mathcal{U}_1] = u_1$. $\pi$ thus also satisfies the constraint that $\mathbb{E}_\pi[\mathcal{U}_2] = u_2 := a \cdot u_1 + b$. Note that any policy satisfying $\mathbb{E}_\pi[\mathcal{U}_2] = u_2$ also satisfies $\mathbb{E}_\pi[\mathcal{U}_1] = u_1$ because the map $x \mapsto a \cdot x + b$ is injective (since $a \neq 0$). Thus $\pi$ must also be a maximum entropy policy satisfying $\mathbb{E}_\pi[\mathcal{U}_2] = u_2$, and so $\pi \in \Pi^{\text{maxent}}_{\mathcal{U}_2, u_2}$ and thus $\pi \in \Pi^{\text{maxent}}_{\mathcal{U}_2}$.

The converse is similar.

$\square$

**Proposition 3.2** (Bounds). *Let $M$ be a CID with utility function $\mathcal{U}$. Then for any policy $\pi$ we have $0 \leq \text{MEG}_\mathcal{U}(\pi) \leq \sum_{D \in \mathcal{D}} \log(|\text{dom}(D)|)$, with equality in the lower bound if $\pi$ is the uniform policy, and equality in the upper bound if $\pi$ is the unique optimal (or anti-optimal) policy with respect to $\mathcal{U}$.*

*Proof.* Recall that

$$\text{MEG}_\mathcal{U}(\pi) = \max_{\pi^{\text{maxent}} \in \Pi^{\text{maxent}}_\mathcal{U}} \mathbb{E}_\pi \left[ \sum_{D \in \boldsymbol{D}} \left( \log \pi^{\text{maxent}}(D \mid \mathbf{Pa}_D) - \log \frac{1}{|\text{dom}(D)|} \right) \right].$$

For the lower bound, first note that the uniform policy $\pi^{\text{unif}}$ is always an element of $\Pi^{\text{maxent}}_\mathcal{U}$, since it is the maximum entropy way to attain a utility of $\mathbb{E}_{\pi^{\text{unif}}}[\mathcal{U}]$. It follows that

$$\text{MEG}_\mathcal{U}(\pi) \geq \mathbb{E}_\pi \left[ \sum_{D \in \boldsymbol{D}} \left( \log \pi^{\text{unif}}(D \mid \mathbf{Pa}_D) - \log \frac{1}{|\text{dom}(D)|} \right) \right] \tag{4}$$

$$= \mathbb{E}_\pi \left[ \sum_{D \in \boldsymbol{D}} \left( \log \frac{1}{|\text{dom}(D)|} - \log \frac{1}{|\text{dom}(D)|} \right) \right] \tag{5}$$

$$= 0, \tag{6}$$

which shows both the lower bound and the fact that we attain it when $\pi = \pi^{\text{unif}}$.

For the upper bound, note that the $\log \pi^{\text{maxent}}(D \mid \mathbf{Pa}_D)$ term is at most 0, so

$$\text{MEG}_\mathcal{U}(\pi) \leq \mathbb{E}_\pi \left[ \sum_{D \in \boldsymbol{D}} \left( 0 - \log \frac{1}{|\text{dom}(D)|} \right) \right] \tag{7}$$

$$= \sum_{D \in \boldsymbol{D}} \log(|\text{dom}(D)|) \tag{8}$$

To see that $\pi$ being a unique optimal (or anti-optimal) policy is a sufficient condition for attaining the upper bound, note that there always exists a deterministic optimal and anti-optimal policy in a CID, so a unique such policy must be deterministic. Further, $\pi$ must be in $\Pi^{\text{maxent}}_\mathcal{U}$, since there is no higher entropy way to get the same expected utility. Then since $\mathbb{E}_\pi[\log \pi^{\text{maxent}}(D \mid \mathbf{Pa}_D)] = 0$, choosing $\pi^{\text{maxent}} = \pi$ obtains the upper bound.

$\square$

**Proposition 3.3** (No goal-directedness without causal influence). *Let $M = (G, P)$ be a CID with utility function $\mathcal{U}$ and decision variables $\boldsymbol{D}$ such that, $\boldsymbol{Desc}(\boldsymbol{D}) \cap \boldsymbol{Pa_U} = \emptyset$. Then $\text{MEG}_\mathcal{U}(\boldsymbol{D}) = 0$.*

*Proof.* Since $\mathcal{U}$ is not a descendant of $\boldsymbol{D}$, it follows from the Markov property of causal Bayesian networks that $\mathcal{U} \perp \boldsymbol{D} \mid \mathbf{Pa}_D$. That means all policies achieve the same expected utility $u$. So, the maximum entropy policy set $\Pi^{\text{maxent}}_\mathcal{U}$ contains only the uniform policy. We get that

$$\text{MEG}_\mathcal{U}(\pi) = -\mathbb{E} \left[ \sum_{D \in \boldsymbol{D}} \log \frac{1}{|\text{dom}(D)|} - \log \frac{1}{|\text{dom}(D)|} \right] = 0. \quad \square$$

**Theorem 4.1** (Pseudo-terminal goals). *Let $M = ((\boldsymbol{V}, \boldsymbol{E}), P)$ be a CBN. Let $\boldsymbol{D}, \boldsymbol{T}, \boldsymbol{S} \subseteq \boldsymbol{V}$ such that $\boldsymbol{D} \perp \boldsymbol{T} \mid \boldsymbol{S}$. Then $\mathrm{MEG}_{\boldsymbol{T}}(\boldsymbol{D}) \leq \mathrm{MEG}_{\boldsymbol{S}}(\boldsymbol{D})$.*

*Proof.* We will show that the maximum entropy policy set $\Pi^{\mathrm{maxent}}_{\boldsymbol{\mathcal{U}^T}}$ (where $\boldsymbol{\mathcal{U}^T}$ is the set of all utility functions over $\boldsymbol{T}$) is a subset of $\Pi^{\mathrm{maxent}}_{\boldsymbol{\mathcal{U}^S}}$, so the maximum predictive accuracy taken over the latter upper bounds the maximum predictive accuracy taken over the former.

Let $\pi \in \Pi^{\mathrm{maxent}}_{\boldsymbol{\mathcal{U}^T}}$, so $\pi = \pi^{\mathrm{maxent}}_{\mathcal{U}, u}$ for some $\mathcal{U}^T \in \boldsymbol{\mathcal{U}^T}$. If we can find a utility function $\mathcal{U}^S \in \boldsymbol{\mathcal{U}^S}$ such that for all $\pi$, $\mathbb{E}_\pi \left[ \mathcal{U}^S \right] = \mathbb{E}_\pi \left[ \mathcal{U}^T \right]$, then a maximum entropy policy with $\mathbb{E}_\pi \left[ \mathcal{U}^T \right] = u$ must also be a maximum entropy policy with $\mathbb{E}_\pi \left[ \mathcal{U}^S \right] = u$. It would follow that $\pi \in \Pi^{\mathrm{maxent}}_{\boldsymbol{\mathcal{U}^T}}$ and so $\Pi^{\mathrm{maxent}}_{\boldsymbol{\mathcal{U}^S}} \subseteq \Pi^{\mathrm{maxent}}_{\boldsymbol{\mathcal{U}^T}}$.

To construct such a utility function, let $\mathcal{U}^S(\boldsymbol{s}) = \sum_t P(\boldsymbol{T} = \boldsymbol{t} \mid \boldsymbol{S} = \boldsymbol{s}) \mathcal{U}^T(\boldsymbol{t})$. Note that since $\boldsymbol{D} \perp \boldsymbol{T} \mid \boldsymbol{S}$, $P(\boldsymbol{T} = \boldsymbol{t} \mid \boldsymbol{S} = \boldsymbol{s})$ is not a function of $\pi$. Then for any $\pi$,

$$
\begin{aligned}
\mathbb{E}_\pi \left[ \mathcal{U}^T \right] &= \sum_t P_\pi(\boldsymbol{t}) \mathcal{U}^T(\boldsymbol{t}) \\
&= \sum_s P_\pi(\boldsymbol{s}) \sum_t P_\pi(\boldsymbol{t} \mid \boldsymbol{s}) \mathcal{U}^T(\boldsymbol{t}) \\
&= \sum_s P_\pi(\boldsymbol{s}) \sum_t P(\boldsymbol{t} \mid \boldsymbol{s}) \mathcal{U}^T(\boldsymbol{t}) \quad (\text{since } \boldsymbol{D} \perp \boldsymbol{T} \mid \boldsymbol{S}) \\
&= \sum_s P_\pi(\boldsymbol{s}) \mathcal{U}^S(\boldsymbol{s}) \\
&= \mathbb{E}_\pi \left[ \mathcal{U}^S \right].
\end{aligned}
$$

$\square$

## C  Proof of Theorem 5.1

**Lemma C.1.** *Let $\pi_{\beta, \mathcal{U}}$ denote a soft-optimal with respect to utility function $\mathcal{U}$, defined as in Theorem 5.1, with rationality parameter $\beta$. Then for any $\alpha \in \mathbb{R}$, we have that $\pi_{\alpha \cdot \beta, \mathcal{U}} = \pi_{\beta, \alpha \cdot \mathcal{U}}$.*

*Proof.* If either $\alpha$ or $\beta$ are equal to 0, both policies are uniform and we are done. Otherwise, write $Q^{\mathrm{soft}}_{\beta, \mathcal{U}}$ for the soft-Q function for $\mathcal{U}$ with rationality parameter $\beta$. We first show that $Q^{\mathrm{soft}}_{\beta, \alpha \cdot \mathcal{U}} = \alpha \cdot Q^{\mathrm{soft}}_{\alpha \cdot \beta, \mathcal{U}}$ by backwards induction.

First, $Q^{\mathrm{soft}}_{\beta, \alpha \cdot \mathcal{U}, n} = \mathbb{E}\left[ \alpha \cdot U_n \mid d_n, \mathbf{pa}_n \right] = \alpha \mathbb{E}\left[ U_n \mid d_n, \mathbf{pa}_n \right] = \alpha \cdot Q^{\mathrm{soft}}_{\alpha \cdot \beta, \mathcal{U}, n}$.

And for $t < n$, assuming the inductive hypothesis holds for $t + 1$,

$$
\begin{aligned}
Q^{\mathrm{soft}}_{\beta, \alpha \cdot \mathcal{U}, t}(d_t \mid \mathbf{pa}_t) &= \mathbb{E}\left[ \alpha \cdot U_t + \frac{1}{\beta} \mathrm{logsumexp}(\beta \cdot Q^{\mathrm{soft}}_{\beta, \alpha \cdot \mathcal{U}, t+1}(\cdot \mid \mathbf{Pa}_{D_{t+1}})) \middle| d_t, \mathbf{pa}_{t+1} \right] \\
&= \alpha \cdot \mathbb{E}\left[ U_t + \frac{1}{\alpha \cdot \beta} \mathrm{logsumexp}(\beta \cdot \alpha \cdot Q^{\mathrm{soft}}_{\alpha \cdot \beta, \mathcal{U}, t+1}(\cdot \mid \mathbf{Pa}_{D_{t+1}})) \middle| d_t, \mathbf{pa}_{t+1} \right] \\
&= \alpha \cdot \mathbb{E}\left[ U_t + \frac{1}{\alpha \cdot \beta} \mathrm{logsumexp}(\alpha \cdot \beta \cdot Q^{\mathrm{soft}}_{\alpha \cdot \beta, \mathcal{U}, t+1}(\cdot \mid \mathbf{Pa}_{D_{t+1}})) \middle| d_t, \mathbf{pa}_{t+1} \right] \\
&= \alpha \cdot Q^{\mathrm{soft}}_{\alpha \cdot \beta, \mathcal{U}, t}.
\end{aligned}
$$

Then, substituting this into the definition of $\pi_{\alpha \cdot \beta, \mathcal{U}}$ we get

$$\pi_{\beta, \alpha \cdot \mathcal{U}} = \text{softmax}\left(\beta \cdot Q^{\text{soft}}_{\beta, \alpha \cdot \mathcal{U}}\right)$$
$$= \text{softmax}\left(\beta \cdot \alpha \cdot Q^{\text{soft}}_{\alpha \cdot \beta, \mathcal{U}}\right)$$
$$= \text{softmax}\left(\alpha \cdot \beta \cdot Q^{\text{soft}}_{\alpha \cdot \beta, \mathcal{U}}\right)$$
$$= \pi_{\alpha \cdot \beta, \mathcal{U}}.$$

$\square$

**Theorem 5.1** (Maximum entropy policy in MDPs). *Let $M = (G, P)$ be an MDP with utility function $\mathcal{U}$, and let $u \in \text{att}(\mathcal{U})$ be an attainable expected utility. Then there exists a unique maximum entropy policy $\pi^{\text{maxent}}_u \in \Pi^{\text{maxent}}_{\mathcal{U}, u}$, and it has the form*

$$\pi^{\text{maxent}}_{u,t}(d_t \mid \boldsymbol{pa}_t) = \pi^{\text{soft}}_{\beta,t}(d_t \mid \boldsymbol{pa}_t) := \begin{cases} \frac{\exp(\beta \cdot Q^{\text{soft}}_{\beta,t}(d_t|\boldsymbol{pa}_t))}{\sum_{d' \in \text{dom}(D_t)} \exp(\beta \cdot Q^{\text{soft}}_{\beta,t}(d'_t|\boldsymbol{pa}_t))}, & \text{if } \beta \neq 0 \\ \pi^{\text{unif}}(d_t \mid \boldsymbol{pa}_t), & \text{if } \beta = 0. \end{cases} \quad (3)$$

*where $\beta = \text{argmax}_{\beta' \in \mathbb{R} \cup \{\infty, -\infty\}} \sum_t \mathbb{E}_\pi \left[\log(\pi^{\text{soft}}_{\beta',t}(d_t \mid \boldsymbol{pa}_t))\right]$.*

*Proof.* In an MDP, the expected utility is a linear function of the policy, so the attainable utility set is a closed interval $\text{att}(\mathcal{U}) = [u_{\min}, u_{\max}]$. We first consider the case where $u \in (u_{\min}, u_{\max})$.

In this case, we are seeking the maximum entropy policy in an MDP with a linear constraint satisfiable by a full support policy (since $u$ is an interior point), so we can invoke Ziebart's result on the form of such policies [Ziebart, 2010, Ziebart et al., 2010, Gleave and Toyer, 2022]. In particular, our feature is the utility $\mathcal{U}$. We get that the maximum entropy policy is a soft-Q policy for a utility function $\beta \cdot \mathcal{U}$ with a rationality parameter of 1, where $\beta = \text{argmax}_{\beta' \in \mathbb{R}} \sum_t \mathbb{E}_\pi \left[\log(\pi^{\text{soft}}_{\beta'}(d_t \mid \boldsymbol{pa}_t))\right]$. By Lemma C.1 this can be restated as a soft-Q policy for $\mathcal{U}$ with a rationality parameter of $\beta$. It follows from Ziebart that $\beta = \text{argmax}_{\beta' \in \mathbb{R}} \pi^{\text{soft}}_{\beta'}$, and allowing $\beta = \infty$ or $-\infty$ does not change the argmax.

In the case where $u \in \{u_{\min}, u_{\max}\}$, it's easy to show that the maximum entropy policy which attains $u$ randomises uniformly between maximal value actions (for $u_{\max}$) or minimal value actions (for $u_{\min}$). These policies can be expressed as $\lim_{\beta \to \infty} \pi^{\text{soft}}_\beta$ and $\lim_{\beta \to -\infty} \pi^{\text{soft}}_\beta$ respectively. $\square$

# D  Experimental Details

## D.1  Tables of results

|         | Known Utility | Unknown Utility |
|---------|---------------|-----------------|
| $k = 1$ | 37.8          | $34.3 \pm 2.6$  |
| $k = 2$ | 21.4          | $32.1 \pm 0.5$  |
| $k = 3$ | 16.8          | $33.6 \pm 0.5$  |
| $k = 4$ | 18.9          | $35.4 \pm 0.6$  |

|                     | Known Utility | Unknown Utility     |
|---------------------|---------------|---------------------|
| $\varepsilon = 0.1$ | 2.4           | $26.1 \pm 0.11$     |
| $\varepsilon = 0.2$ | 1.5           | $17.4 \pm 0.2$      |
| $\varepsilon = 0.3$ | 0.95          | $11.0 \pm 0.25$     |
| $\varepsilon = 0.4$ | 0.50          | $6.2 \pm 0.08$      |
| $\varepsilon = 0.5$ | 0.20          | $2.9 \pm 0.06$      |
| $\varepsilon = 0.6$ | 0.04          | $1.0 \pm 0.003$     |
| $\varepsilon = 0.7$ | 0.003         | $0.10 \pm 0.002$    |
| $\varepsilon = 0.8$ | 0.001         | $0.10 \pm 0.003$    |
| $\varepsilon = 0.9$ | 0.008         | $0.091 \pm 0.007$   |

i

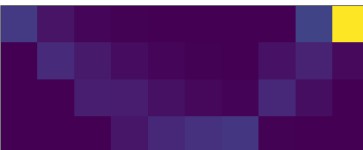
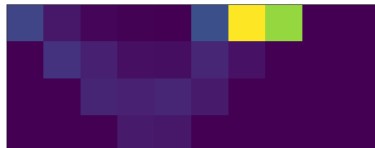

(a) Occupancy measures of optimal policy when $k = 1$.

(b) Occupancy measures of optimal policy when $k = 4$.

Figure 5: Occupancy measures

## D.2 Visualising optimal policies for different lengths of goal region.

Figure 5a and Figure 5b show the occupancy measures for an optimal policy for k=1 and k=4 respectively, where $k$ is the length of the goal region in squares. Although the goal region is larger in the latter case, the optimal policy consistently aims for the same sub-region. This explains why unknown-utility MEG is higher than MEG with respect to the environment reward. The policy does just as well on a utility function whose goal-region is limited to the nearer goal squares as it does on the environment reward, but fewer policies do well on this utility function, so doing well on it constitutes more evidence for goal-directedness.

## D.3 Further details

The experiments were carried out on a personal laptop with the following specs:

- *Hardware model:* LENOVO20N2000RUK
- *Processor:* Intel(R) Core(TM) i7-8665U CPU @ 1.90GHz, 2112 Mhz, 4 Core(s), 8 Logical Processor(s)
- *Memory:* 24.0 GB

We used an environment from the SEALS library[8], and adapted an algorithm from the imitation library[9]. Both are released under the MIT license.

For information on hyperparameters, see the code.

---

[8]https://github.com/HumanCompatibleAI/seals
[9]https://github.com/HumanCompatibleAI/imitation

