# OpenReview forum: "Measuring Goal-Directedness"
_NeurIPS.cc/2024/Conference — NeurIPS 2024 spotlight_

### Official Review · Reviewer_Kmdz · 2024-07-07

**Soundness:** 4
**Presentation:** 3
**Contribution:** 2
**Rating:** 6
**Confidence:** 3

**Summary:**

The paper proposes a measure of evidence for goal-directedness based on the maximum causal entropy principle, MEG. It is operationalized as the ability to predict variable D based on the hypothesis that it is optimizing a utility function U, where U represents the goal and D the agent’s decisions. Specifically, they try to predict the agents' decisions using the maximum-entropy policy that achieves the same utility as the agent’s own policy. MEG is essentially the negative cross-entropy of this prediction. The authors also describe algorithms for estimating MEG, and demonstrate it in action with some small-scale experiments.

**Strengths:**

- The work is well-motivated and situated among prior work

- The writing is clear and the examples provided are very helpful for understanding

- The MEG satisfies some nice properties such as invariance to translation and scaling

- The authors allow MEG to cover a wide range of use cases, for instance

  - It can be estimated even when only a set of trajectories is available, rather than the agent’s policy itself.

  - It can be extended to settings where the the utility function is unknown

**Weaknesses:**

- It is unclear how useful MEG will be for checking goal-directedness of generalist AI agents acting in the real world, because:
  - it is expensive to compute, since it requires optimizing the maximum entropy policy, and their estimation algorithm can only guarantee a lower bound on MEG for the unknown-utility case
  - It requires a causal model containing the target variables that the agent might be trying to manipulate.

- The name and framing of MEG is slightly misleading: MEG is not directly a measure of how goal-directed an agent is, but rather the amount of evidence for goal-directedness. The maximum value MEG can take varies based on the number of possible choices in an environment. Thus, it may be difficult to compare the goal-directedness of different agents’ behavior if we could only observe them acting in different environments. (e.g., to determine if agents acting in our constantly changing world are becoming increasingly goal-directed)

- typo: agentcy on line 18

- Capitalization of v at the end of definition 2.1 is inconsistent

**Questions:**

Why does the maximizing policy in equation 1 always obtain the same expected utility as pi?

**Limitations:**

yes

---

> ### Author Rebuttal · Authors · 2024-08-07
>
> Thanks for the review.
>
> **Lower bound.**
> It’s true that our algorithm can only guarantee a lower bound in the unknown-utility case, but this is true whenever one performs SGD on a nonconvex loss function to estimate some quantity. With neural networks, we often get good estimates of the quantity regardless. We’ve added a note to clarify.
>
> **Misleading name.**
> It’s true that MEG only measures the evidence for goal-directedness exhibited in the behaviour (with respect to the variables we measure), not necessarily the full goal-directedness of the agent. But this is generally true of tests: a high-school maths test cannot measure the full ability of a star mathematician, since there is limited opportunity to provide evidence for it while completing the test. Similarly, depending on the environment in which we measure MEG, we may see relatively little evidence for an agent’s goal-directedness. But we think it’s still fair to say that MEG is measuring goal-directedness, just as the test is measuring mathematical ability.
>
> We should of course bear this point in behind, and think of MEG as measuring the goal-directedness with respect to a given environment distribution. We have updated the paper to clarify this.
>
>
>
> **Equation 1.**
> The fact that the maximising policy obtains the same expected utility as pi is non-obvious but follows from Ziebart’s result. I’m not sure why we didn’t include a proof of this – we have included one now. Thanks for drawing our attention to it.

---

> > ### Comment · Reviewer_Kmdz · 2024-08-08
> > **thank you for the response**
> >
> > **Lower bound**  "we often get good estimates of the quantity regardless" this is a bit too vague to assuage my concerns. "Good" might not be useful, depending on how the metric is being used: e.g, if we are trying to determine which of agents A and B is more goal-directed, we might obtain a higher bound for agent A than agent B, even if agent B is in fact more goal directed than agent A.
> >
> > I am satisfied with the other replies, thank you for clarifying.

---

> > > ### Author Response · Authors · 2024-08-14
> > > **Agreed**
> > >
> > > You are right that this is an important consideration. In some cases we can find MEG via global optimisation or tabular methods, and a good idea for future work is to compare the results of such methods against SGD in these cases.

---

### Official Review · Reviewer_8w2P · 2024-07-11

**Soundness:** 3
**Presentation:** 2
**Contribution:** 4
**Rating:** 8
**Confidence:** 3

**Summary:**

* The work introduces a framework for quantifying the goal-directedness of a
  decision-making system with respect to a particular utility function or a
  family of utility functions.
* The method is grounded in Dennett's *intentional stance* and Jaynes'
  *maximum entropy inference*. The authors have made some additional choices
  while formalising these principles to arrive at their formulation, which
  ends up similar to Ziebart's Maximum Causal Entropy IRL framework.
  The resulting formula is essentially a measure of how close the subject
  policy comes, in terms of cross entropy, to a maximum entropy formula for
  the utility function at some inverse temperature.
* The authors establish that the resulting formula, MEG, satisfies desirable
  properties for a measure of goal-directedness, namely being invariant to
  the scaling and translation of utility functions, having interpretable
  upper and lower bound policies for a given utility function, and measuring
  only causal goal-directedness.
* The authors extend their measure of goal-directedness with respect to a
  single utility function into an aggregate measure of goal-directedness with
  respect to a family of utility functions (simply by taking the maximum MEG
  over the family of utility functions). This also allows them to express a
  measure of goal-directedness with respect to a set of 'target variables',
  by considering the family of utility functions defined over those
  variables.
* The authors derive algorithms for computing both kinds of MEG in the
  setting of causal influence models derived from MDPs. The algorithms are
  based on gradient-based search requiring repeatedly computing maximum
  entropy policies for various inverse temperatures (and, in the case of a
  family of utility functions, varying family parameters).
* The authors implement their algorithm for some simple MDPs as a 'proof of
  concept' experiment, demonstrating how various factors influence the
  measure of goal-directedness.

**Strengths:**

I thank the authors for submitting their valuable work which is in a position to make an excellent contribution to the field.

* I think the goals of the work are exceptionally well-motivated. The authors
  did not spend much time elaborating on the need for measures of goal
  directedness, so I note that such measures have immediate applications in
  attempts to rigorously study risks from learned optimisation in deep
  learning. Emergent goal-directedness is a core component of the problem of
  AI safety (for modern systems and future more capable systems) and scalable
  methods for quantifying goal-directedness are sorely needed by researchers
  such as myself who want to better understand this phenomenon in order to
  reduce or avert these risks.
* The authors have taken careful steps to justify their proposed framework
  for measuring goal-directedness, including showing that it satisfies some
  sensible properties and showing that in simple examples (including toy
  experiments with some MDPs) the values given by the MEG measure are
  sensible.
* Overall, the paper is fairly clearly written and I feel I was able to
  understand what the authors have to say. I think that this strength is
  limited in that there may be some room for improvement in the flow of
  sections and in some low-level details of the writing to make the paper
  optimally clear and accessible (see below), however overall it was
  passable.

**Weaknesses:**

I noted the following issues with the work, based on my understanding.

1.  MEG is described as "philosophically grounded" because of its roots in an
    interpretation of Dennett's Intentional Stance. However, it does not
    appear to be a unique interpretation and this philosophy does not appear
    to be the only possible philosophical approach to defining agency.

    * Related works Kenton et al. and Orseau et al. both appeal to Dennett's
      philosophy but arrive at different formalisms. The authors have not
      established that their interpretation is uniquely philosophically
      grounded. (Appendix A begins to address this with respect to Kenton et
      al.'s prior work, but it would be good to see a discussion of the
      relationship with Orseau et al.'s approach.)

    * If MEG were a unique formalisation of Dennett's philosophy, it would
      still appear to be a behaviourist approach, which stands in contrast to
      a mechanistic approach that looks into the internals of a system for
      explicit representations of goals and for the implementation of
      decision-making processes.

2.  I am concerned that the statements of proposition 3.1 and 3.2
    are incorrect as stated.

    * **Proposition 3.1.** There must be some missing assumption in this
      theorem statement that $a$ is a positive scalar. Otherwise you have
      proven that a MEG policy for any given utility function is also
      goal-directed towards a negated version of that policy. Could you
      please clarify whether this is the case or whether I am missing
      something that means that a policy that pursues the negative of a
      utility function is somehow goal-directed towards that utility
      function?

    * **Proposition 3.2.** It is well known that in many examples there are
      decisions for which multiple actions lead to equal expected utility and
      are therefore equally optimal. In such cases there is no unique optimal
      policy. What happens to the bounds in this case? The statement of the
      proposition would appear to be ill-defined in case there is no unique
      optimal policy. Is it the case that you do not derive an upper bound in
      this case? The distinction being then that there still will be some
      upper bound (perhaps?) but that you have not characterised it?

    When I checked the proofs for these propositions in the appendix I found
    them to provide insufficient detail to resolve my concerns or point to a
    particular mistake in the reasoning.

    I am recommending acceptance of the paper on the assumption that these issues will be easy to dismiss or to correct.

3.  In the unknown-utility case, MEG fails to account for complexity of
    utility functions within the hypothesis class.

    * It is well known in RL that for any policy, a utility function exists
      that can rationalise that policy. For example, there is a reward
      function that immediately, highly rewards whichever actions that policy
      happens to take in each state.
    * This leads me to wonder if for any policy it is possible to find some
      reward function for which MEG very strongly indicates that this policy
      is goal-directed towards that utility function.
    * Especially in the case of MEG for unknown utility functions, it is
      therefore possibly that if the hypothesis class of utility functions is
      too rich, the MEG for unknown utility functions will register almost
      any policy as goal-directed.
    * I think something that is missing is some way of accounting for the
      complexity of utility functions. Philosphically, a complex utility
      function is not 'useful for predicting behaviour' if it is itself as
      complex as the behaviour, even if it is capable of predicting behaviour
      well.
    * I think a significant weakness of MEG, especially in the
      unknown-utility setting, is that it leaves the user of the framework to
      rule out too-complex utility functions by the coarse tool of including
      them or excluding them from the hypothesis class, rather than providing
      a principled tool for managing this complexity.

    If the authors are aware of this issue but consider addressing it to be
    out of scope, I would like to see it acknowledged as such in the main
    text.

4.  If I am not mistaken, the scalability of the method for computing MEG
    relies on solving an RL problem (computing a maximum entropy policy in
    MDPs). If this is the case then I think the scalability of the method is
    questionable.

    * It is certainly common practice to solve RL problems, and in any case
      where we can train a policy that will be the target of MEG evaluation,
      we can presumably also train a max-ent policy to measure its MEG.
      Perhaps calculating such a policy repeatedly will be expensive but it
      should at least be feasible to do so.
    * However, the very applications I think motivate this work are settings
      where we don't understand the internal motivational system of the
      trained system (and so MEG can help quantify that). This would of
      course only happen if we are training the subject policy in such a
      complex environment that its emergent goal-directedness is in question,
      due to, for example, potential goal misgeneralisation (or so-called
      'inner alignment failure').
    * In such a setting, I suppose we would share the same concerns about the
      max-ent policy required for quantifying MEG. If I am not mistaken, if
      this probe policy's generalisation could not be trusted, then neither
      could we trust the result of using it to calculate the MEG of the
      subject policy.

    There is a comment in the discussion to the effect that future work can
    consider the setting of 'distributional shifts' which is normally
    considered to be a prerequisite for the phenomenon of goal
    misgeneralisation. However, I think this concern undercuts the motivation
    of the present work in its early stages and if there is no feasible way
    to reliably compute the MEG for complex learned systems where the
    internal goals are in question then the work may not achieve its goals in
    the future, and this issue should be raised and (ideally) addressed now.

5.  There is insufficient explanation of relationship to prior work, particularly to IRL and agents and devices.

    * **Inverse reinforcement learning.** IRL represents a large body of
      work including theoretical work aimed at finding goals that
      rationalise particular policies, which is clearly closely related to
      MEG. The authors' brief comments in the related work section have not
      clarified to me the relationship between MEG and IRL.
      In particular, it seems plausible to me that MEG is essentially
      functions similar to a subcomponent of a IRL framework based on
      Bayesian inference.

        * A Bayesian approach for IRL for example involves positing a
          likelihood of the behavioural data given each goal. This likelihood
          function represents a kind of measure of how well the behaviour can
          be explained by that goal. This seems similar to MEG, though of
          course MEG does not adopt a Bayesian framework.
        * Similarly, what is the relationship between unknown-utility MEG
          with a given hypothesis class and the model evidence in a Bayesian
          formulation of IRL with the same hypothesis class? Of course there
          is a difference of integrating versus taking a max, but my point is
          that these are different approaches to the same kind of problem.

        I think the paper would be strengthened if the authors could clarify
        whether MEG is addressing a fundamentally new kind of problem or one
        that arises naturally in the course of IRL. If it's not new, that
        seems fine, the approach is apparently new---but it would be nice to
        see a discussion of how the approach compares to the Bayesian
        approach outlined above.

    * **Agents and devices.** This prior works appears to present a
      Bayesian framework for distinguishing agents from non-agents, along
      with one particular instantiation of this general framework in terms
      of epsilon-greedy suboptimality and Solomonoff induction.
      The comments in the related work section appear to relate to the
      instantiation. What is the relationship between the present work and
      the general framework from Orseau et al.?
      For example, would it be possible to derive a maximum-entropy version
      of Orseau et al.'s framework, and if so, how might that compare to
      MEG? Does the difference come down to the difference between Bayesian
      inference and maximum entropy inference?

    I would have liked to see sections along the lines of appendix A (for
    Kenton et al.) drawing out these relationships.

If the authors are able to address each of these issues to my satisfaction I think my assessment of the paper's strength would be greatly improved and I would be willing to substantially strengthen my recommendation to accept the paper.

**Questions:**

Many of the weaknesses I raised above are partially framed as questions since
I am not fully confident that I have understood the work in forming these
comments (I welcome corrections from the authors).

Here I collect additional questions that are less crucial to the overall
validity of the work but are still either barriers to my understanding or
seem like minor issues in the presentation.

* The introduction claims that "pursuing goals is about having a particular
  causal effect on the environment." This appears to be part of the case for
  MEG being "philosophically grounded".
  I can see how in many cases this correspondence holds, but it is not
  immediately obvious to me why there couldn't be cases where the pursuit of
  goals is separated from causality. For example, if a goal is always going
  to be achieved counterfactually regardless of an agent's actions, then it
  becomes possible to 'pursue' this goal without having any causal effect on
  its achievement.
  Do the authors have some further justification for this framing, or a
  citation that gives such justification?

* Equation 1:

    * Can the definition of MEG be interpreted as a minimum KL divergence
      between the policy and the closest maximum-entropy policy?
    * The cross entropy between two distributions is not symmetric. Why is is
      justified to take the cross entropy in this particular direction?

* What is the relationship between theorem 5.1 and prior work on maximum
  entropy RL such as Haarnoja et al. 2017 "Reinforcement learning with deep
  energy-based policies"?

* In the discussion section, under "societal implications", there is a
  passage "the relationship between goal-directedness and danger is fairly
  indirect".

    * I wonder, did the authors mean to write "the relationship between
      *measuring* goal-directedness and danger is fairly indirect"?
    * If not, could the authors please elaborate on this indirectness, since it
      seems that the goal-directedness itself is fairly tightly and directly
      connected to risks (as outlined in the works cited in the introduction).


Not a question, but while reading the paper I happened to note the following
minor presentation issues the authors may like to address in future versions.

* Line 5: I think the authors mean "adaptation" rather than "adaption".
* Line 14: The word "mundane" seems to downplay the real and potentially
  devastating impacts discussed in Chan et al. Consider a different word
  or a different angle of contrast (perhaps the scale of the harms?).
* Line 16: Shavit et al. citation and reference missing a date.
* Line 18: "Agentcy".
* Line 22: Incomplete sentence, consider "According to Dennett... [1989],
  *we are justified in doing so whenever it* is useful ...".
* Line 31: MEG acronym was introduced in the abstract is undefined in the
  scope of the introduction.
* Line 65: Orseau et al. should be cited in text mode.
* Line 78: Something appears to have allowed the bottom margin on page 2
  to shrink, causing the final lines on the page to come too close to the
  page number.
* Algorithm 1 line 7: Until convergence of what? Not beta, for example.
* Line 294: This and the next paragraph headings are missing periods on
  the abbreviation "vs.".
* Line 296: "meg" should be capitalised.
* Footnote 1 is missing a period, at the footnote mark (line 296) AND at
  the end of the footnote text.
* Line 300: Potential missing word, "optimal *policies*"?

**Limitations:**

It would have been good to see a more acknowledgement of the limitations of
the work (such as those I have enumerated above in the weaknesses section, pending the authors' clarification), throughout the paper or in a dedicated section of the paper.

Some additional points on addressing these limitations in the paper:

* You could more carefully justify or qualify the discussion of the extent to which the
  approach is uniquely principled / philosophically motivated.

    For example, in the conclusion, MEG is described as "grounded in the
    philosophical literature" when the authors appear to just mean "grounded in
    the philosophy of Dennett" (as they do qualify elsewhere), the difference between which I outline in Weakness (1).

* The scalability of the work does appear to depend on repeatedly computing a
  maximum entropy policy, which would appear potentially many times more
  expensive than computing the subject policy, though perhaps feasible. This
  is based on my understanding. I would have liked to see the authors clarify
  the requirements in comparison to the cost of finding the target policy
  explicitly, since they claim their method is "scalable".

See also weaknesses (3) and (4).

---

> ### Author Rebuttal · Authors · 2024-08-06
>
> We appreciate your extensive and careful review.
>
> **Motivation.**
> Agreed emergent goal-directedness is a core AI safety problem. Updated to be more explicit.
>
> **Terminology.**
> We agree our approach is not a unique interpretation of Dennett, and his view not the only one in the literature. We have changed  "the literature" to  "Dennett". Would “philosophically motivated” be preferable to “philosophically grounded”?
>
> **Mechanistic vs behavioural.**
> Agreed our approach is behavioural rather than mechanistic. We think both approaches should be pursued.  Behavioural approaches seem more tractable, but mechanistic approaches will generalise better off-distribution. We have added this point to the discussion.
>
> **Prop 3.1.**
> The proposition is correct – MEG considers an agent goal-directed wrt a utility function both if it maximises it and if it minimises it. We have made the proof more explicit.
>
> We admit this may be unintuitive. Arguably trying to pessimise a utility function is just as goal-directed as trying to optimise it. But on the other hand, perhaps MEG should be signed. We did consider this. Would the reviewer argue in favour?
>
> **Prop 3.2.**
> The proposition is correct, but was stated ambiguously. We have updated it to say, “there is equality in the upper bound if and only if pi is a *unique* optimal (resp. anti-optimal) policy”. Is the meaning now clear?
>
> **Is unknown utility meg always high?**
> There does alway exists a reward function to rationalise a (deterministic) policy. But this only applies to state-action or state-action-state rewards, not state rewards (ignoring the constant reward function). It’s the same with MEG: all (deterministic) policies are maximally goal-directed with respect to utility functions over <observation,action> pairs. But if the set of functions if over some downstream variable (e.g. states in an MDP), most policies will not be maximally goal-directed. We think MEG gets this right -- it's hard to say an agent is not optimising its own behaviour, but its easy to say whether it's optimising some variable out in the world.
>
> **Complexity of utility functions.**
>
> Any way of measuring goal-directedness needs a way to avoid the trivial result that all policies are maximally goal-directed. One option is indeed to penalise more complex utility functions. But MEG does it in a different way: by considering utility functions over specific sets of variables. If the variables considered don’t include the agent’s actions, then most policies will not be very goal-directed, for any complexity of utility function. So we think it’s best to present MEG without an additional complexity penalty, although we acknowledge that adding one may be natural. We have now mentioned this in the paper.
>
> **Scalability.**
> MEG does require repeatedly solving an RL problem. MEG is about as scalable as MCE IRL, so we expect it to scale better than UTM-based approaches (Agents and Devices). But we have not demonstrated the scalability here, so we have now weakened the language. We agree that much usefulness depends on scaling. However, there is also value in conceptual clarification, and taking a step towards scalable methods.
>
> **Generalisation.**
> Agreed distributional shifts are a key motivation. Rather than trusting the imitation policy to generalise correctly, a better approach could be to think of MEG as measuring goal-directedness on a given distribution, and to proactively seek out distributions to test on, e.g. where unknown-utility MEG is much higher than intended-reward MEG. This may let us detect goal misgeneralisation.
>
> **Prior work.**
> We have added new sections in the appendix comparing to A&D and IRL. One key difference with IRL: IRL does not try to tell you *how* goal-directed a policy is, just what its goal is. Consider an agent that performs well on the intended reward by default, but badly under a distributional shift. IRL would just tell you that the utility function of the agent changes, but MEG can tell us whether the behaviour is strongly goal-directed on the new distribution, i.e. whether we have capability misgeneralisation (low MEG) or goal misgeneralisation (high MEG).
>
> **Does pursuing goals require a causal effect?**
> This is a common view in philosophy (see section 3.1 of the SEP article on agency). But it’s not universal, e.g. *evidential decision theory* is concerned with pursuing goals even if one cannot causally affect them. We think that MEG can be generalised to an “evidential” version, but it’s beyond the scope. We have added a reference and a footnote.
>
> **KL div, cross entropy, Haarnoja.**
> - MEG is similar to the minimum KL with a maxent policy. You would remove the second term (the entropy of the uniform policy) and add a term for the negative entropy of the policy itself. The first change is just a translation. The second would mean that a less entropic way of achieving a certain utility would be considered more goal-directed than a more entropic way.
>
> - The direction of the cross entropy gets us a maxent policy that well-predicts the agent’s policy, rather than vice versa. Reversing it would mean maximising the maxent policy's probability of the agent's *highest probability* action, i.e. ignoring other aspects of the policy.
>
> - Haarnoja’s results are essentially a useful alternative formulation of Ziebart’s. We could probably reformulate our results along the lines of Haarnoja’s, but chose to focus on Ziebart.
>
> **Harm.**
> We agree the relationship is not indirect. Probably we meant to include the word “measuring”, but this too now seems like an overstatement. We have removed the sentence.
>
> **Limitations.**
> We have added the following to the limitations section: scalability; mechanistic vs behavioural approaches; more on distributional shifts. Has the rest been satisfactorily clarified?
>
> Let us know if our response is enough to increase your score, or whether further changes would be needed. Thanks again.

---

> > ### Comment · Reviewer_8w2P · 2024-08-08
> > **Thanks, your rebuttal has addressed my concerns, with notes**
> >
> > The revisions proposed by the authors in this rebuttal and their other
> > rebuttals will substantially improve the clarity and quality of the paper.
> > They will also address all of my concerns (subject to the below notes).
> > Accordingly I am pleased to strengthen my recommendation for the paper's
> > acceptance.
> >
> > **Notes on terminology:**
> > Once the authors have clarified that the main philosophical grounding is in
> > Dennett's work, I am satisfied and I leave it to the authors to decide the
> > best choice of wording between "philisophically motivated" and
> > "philosophically grounded".
> >
> > **Notes on proposition 3.1, and goal-directedness of minimisation:**
> > Thank you for clarifying this point about the unsigned nature of MEG, which
> > I missed in my initial review.
> > I think my confusion stemmed from the fact that I picked up the phrasing "a
> > policy pi is goal-directed *towards* a utility function U".
> > The phrasing "directed towards" suggests to me that the policy is acting to
> > maximise the utility function (to some extent), rather than to minimise it.
> > I note that the authors switch between this phrasing and the sign-neutral
> > phrase "goal-directed with respect to" in the paper, but I missed this
> > distinction.
> > I am satisfied with the following resolution:
> >
> > 1. If they have not already done so, the authors should make sure they
> >    reserve "towards" for cases where the policy is positively directed with
> >    respect to the utility function.
> >
> > 2. The authors should also explicitly clarify this aspect of their
> >    framework prominently in the main text.
> >
> > Moreover, the authors also asked if I would argue in favour of a signed
> > measure of goal-directedness. I think I would, but it is not crucial to my
> > evaluation of the present work, any substantial revision in this direction
> > would require further review. Seems like it is best left for future work.
> >
> > **Notes on proposition 3.2, about multiple (anti-)optimal policies:**
> > Thank you for clarifying. I now understand the proposition statement. The
> > wording change you have proposed in the initial rebuttal is sufficient in my
> > judgement.
> >
> > **Notes on distributional shift and goal misgeneralisation:**
> > I appreciate clarification on the proposed application of MEG for use in
> > detecting goal misgeneralisation. I think this is an interesting direction
> > for future work and I am pleased that the authors are expanding their
> > discussion in their revision.
> > I maintain my concerns about the specific proposed methods, since in practice
> > it may not be straightforward to identify novel state distributions or their
> > corresponding intended reward functions for measuring intended-reward MEG.
> > However, this issue alone does not undermine my judgement of this paper as
> > an important theoretical contribution to the field.

---

> > > ### Author Response · Authors · 2024-08-12
> > > **Prop 3.1.**
> > >
> > > Thanks, we have followed your suggestions re: Prop 3.1.

---

> ### Author Response · Authors · 2024-08-07
> **Further Clarification of Prop 3.2**
>
> Just to further clarify Proposition 3.2, as it may still not be clear. The upper bound holds whether or not there exists a unique optimal (or anti-optimal) policy, but iff *there exists* such a policy, *and* pi is that policy, then there is equality in the upper bound. Do you think we should spell this out more explicitly in the statement?

---

### Official Review · Reviewer_AseF · 2024-07-12

**Soundness:** 3
**Presentation:** 3
**Contribution:** 3
**Rating:** 6
**Confidence:** 2

**Summary:**

The paper introduces maximum entropy goal-directedness (MEG), a measure of how much an agent is goal directed towards a given utility function. The authors extend the theoretical framework introduced to the case where the utility function that is being optimized is not known. Moreover, they propose an algorithm to measure MEG in Markov Decision Processes.
To show the practicality of the metric introduced, the authors show the performances of the algorithms empirically. Specifically, they show how MEG relates with the optimality of a $\epsilon$-greedy policy, and how it relates to task difficulty.

**Strengths:**

- The paper is theoretically sound, and well presented
- I found the images and explicit examples to be particularly helpful in understanding how MEG works and why it was defined in a specific way
- Section 4 addresses an important problem, and I found that it provides clear results which are also well explained.
- I have not seen much similar work, and I definitely think it addresses an important problem, especially when talking about future goal-directed LLM systems.

**Weaknesses:**

- As mentioned by the authors, MEG relies on having full access to the causal model of the environment. This can be a highly-demanding assumption in many real environments.
- Given the restrictive assumption mentioned above, the paper should at least investigate in more detail how computationally expensive is computing MEG in large-scale setting.
- The experiments are limited to few toy examples

**Questions:**

- How do you think one could make MEG more robust towards distributional shifts?

- Do you have any example in mind on how MEG could fail when considering how goal-directed an agent is? I think it could be an important point to discuss to give a better perspective to the reader on when one would want to employ such metric. Maybe you have some toy examples which can illustrate this?

- Is it possible for two similar policies to achieve drastic differences in MEG? Do you have any experiments which shows how robust of a metric it is?

-How one could extend MEG to cases where the causal environment is only partially known?

- Do you have some experiments/comparison/thoughts on how MEG relates directly with agency?

**Limitations:**

no ethical limitations

---

> ### Author Rebuttal · Authors · 2024-08-07
>
> Thanks for the review.
>
> **Requiring a causal model**.
> A full causal model is not strictly necessary for computing MEG. What is needed is a simulator where you can measure the variables of interest under different policies. We have updated the paper to make this clearer. That said, investigating how MEG can be applied in practice is indeed an important direction for future work.
>
> **Robustness to distributional shifts**.
> One simple idea is to proactively seek out distributions where MEG is especially high, either by hand or by using MEG as an optimisation target. It would be especially interesting to find distributions such that goal-directedness with respect to the intended reward function is low, but goal-directedness with respect to some other reward function is high. This could be a way to detect goal misgeneralisation (https://arxiv.org/abs/2105.141110).
>
> **How MEG can fail.**
> One failure mode is that if you measure goal-directedness with respect to the agent’s own actions (or more correctly, state-action pairs in an MDP, or observation-action pairs in a CID), then the agent will appear maximally goal-directed regardless of its policy. This means that if you measure the goal-directedness of an LLM with respect to the text it generates, for example, it won’t tell you much. Instead you need to measure the goal-directedness with respect to some downstream variable (such as a user’s response, or the outcome of a task). We have updated the appendix to mention this example.
>
> **Can similar policies get very different MEG?**.
> Yes, at least under some reasonable notions of policy-similarity. Consider an example where two policies in an MDP agree in every state *except the start state*, where one policy takes action a1 and the other takes a2. If a1 leads to a part of the state space where the policies are both very goal-directed, and a2 leads to one where they are not, then there can be an arbitrarily large difference in goal-directedness, despite the policies agreeing in almost every state. This underlines that MEG measures the goal-directedness on a particular distribution. We have added this example to the appendix.
>
> **The case where the causal model is only partially known.**
> As mentioned above, we can tackle this case as long as we can simulate the environment under different policies. The simulation may only partially specify a causal model, as it may not allow interventions on non-policy variables. It may be interesting to explore other forms of partial knowledge.
>
> **How MEG relates to agency.**
> Agency is an overloaded term with a range of meanings and connotations. Goal-directedness is often considered a key component of agency. For example, Dennett's intentional stance only applies to systems pursuing goals in some sense, and Dung (https://philarchive.org/rec/DUNUAA) argues that it underlies other aspects of agency such as autonomy, empowerment, and intentionality. Some of these notions are still vague, but it's an interesting line of future research to try to make these other aspects precise, and formally and/or theoretically relate them to the notion of goal-directedness we present here.

---

> > ### Comment · Reviewer_AseF · 2024-08-08
> > **Thanks for the clarifications**
> >
> > Dear authors,
> > Thanks for your answer.
> >
> > **Requiring a causal model**: This addresses one important critique and concern I had. Thanks.
> >
> > **Distributional Shift**: I agree that this could be a sensible way of detecting goal misgeneralisation. I would especially be curious to know if MEG could be used to identify also the 'cause' of themisgeneralisation, and not only if this is happening or not.  This could be done by considering increasingly subsets of the Causal Model and see which of the downstream variable causes the shift in MEG. Obviously this goes beyond the scope of this work, but thanks for clarifying my question
> >
> > **How MEG can fail** and **Can similar policies get very different MEG**: These examples are clear to me, and I appreciate that they have been added to the appendix. I would like the authors to clarify in the final version that *MEG measures the goal-directedness on a particular distribution*
> >
> > **The case where the causal model is only partially known**and **How MEG relates to agency**: Thanks for your answer.
> >
> >
> > To summarize, all the issues I raised were addressed by the authors. I will increase the score accordingly.

---

### Official Review · Reviewer_8tgY · 2024-07-12

**Soundness:** 4
**Presentation:** 4
**Contribution:** 4
**Rating:** 7
**Confidence:** 3

**Summary:**

The paper is studying a problem that is both mathematical and philosophical in nature: how can we measure quantitatively, whether an observed behaviour is goal-directed or not (or to what extent it is). It tackles this problem by starting from a causal model of the world, where there are explicit utility variables, state variables, and decision variables. For a known utility function, it proposes to measure goal-directedness by essentially asking: given an observed policy $\pi$, find a policy that is maximizing the utility function but regularised by entropy that is _most predictive_ of $\pi$. Compare its predictive accuracy to uniform prediction, and this quantifies goal-directedness. They also extend this to the case of unknown utilities, by considering parameterized sets of utility functions. The favourable properties of this measure is discussed, and its practicality is demonstrated with a simple empirical result on CliffWorld.

**Strengths:**

- The question of measuring goal-directedness is tremendously significant, since it is a core requirement for agency.
- The proposed solution is sufficiently novel. Even though it builds on the Maximum Causal Entropy IRL, it sufficiently extends it and applies it in a completely new way.
- The paper is written very clearly and it is easy to follow.

**Weaknesses:**

- Societal implications could have been discussed in more detail, especially how this "could enable bad actors to create even more dangerous systems." An extended discussion about this added to the appendix would improve this.

- A more comprehensive empirical demonstration of the practicality would improve the paper.

**Questions:**

Q: I have one question about the unknown utility setting. If I get this right, in general, if the set of possible utility functions is large enough, and the causal model is rich enough, I can potentially find a utility function making many behaviours goal-directed, even though they may not be. For example, consider the behaviour of a river, as its flow. If my causal model represents enough things relevant (e.g. elevation, geography, physics, weather etc.), I can potentially find a utility function in the lines of _minimize travel distance to the nearest sea/lake_ where elevation incurs a cost, and conclude that the flow of the river is goal-directed. Is this correct? If so, it seems like one must be careful in terms of the class of utility functions they consider.

Bonus Q: I know that the paper assumes a known causal model. However, I am curious. How would we deal with hidden confounders here? Is there no hope? Specifically, could there be reasons internal to the agent that make a seemingly non-goal-directed behaviour in fact goal-directed, unbeknownst to us?

**Limitations:**

Paper discusses limitations sufficiently.

---

> ### Author Rebuttal · Authors · 2024-08-06
>
> Thanks for your review. We agree that the problem of measuring goal-directedness is of great significance and we’re glad you feel we have made a novel contribution here.
>
> **Societal implications**.
> We agree this should be discussed in more detail. Thanks for the suggestion of an extended discussion in the appendix about the concern of bad actors using our ideas to create more dangerous systems. We have added a section in which we weigh various considerations, and then explain why we ultimately think it’s not a major concern.
>
> **Empirical demonstration**.
> The contribution of this paper is primarily theoretical and conceptual, and so we decided to include experiments mostly for illustrative purposes. Investigating scalability, robustness, and practical applications (like detecting goal misgeneralisation) are promising directions for future work.
>
> **Question about the unknown utility setting.**
> Good question. Every system you model will be goal-directed with respect to a utility function over its own actions, but beyond that, it’s not the case that by including enough variables in your causal model, you can always find a utility function towards which the system is goal-directed. In fact, we can apply Theorem 4.1 to this question, and say that if the agent is not very goal-directed with respect to some Markov blanket of variables surrounding its action, then it’s no more goal-directed with respect to any variable further away in the causal model.
>
> Your river example is an interesting one. Depending on how you model it, it may indeed be the case that the river appears quite goal-directed. If we think of the river as being able to change to flow in any direction at various points along its path, but choosing to follow the path it does, then yes, it will have high goal-directedness with respect to its travel distance to the sea, as you suggest. This is in line with famous examples of very simple agents, such as a thermostat.
>
> On the other hand, it’s debatable that you should model the river as “being able to” flow in any direction. In general, the smaller the set of actions the system has available, the lower its maximum goal-directedness, as we show in Proposition 3.2.  Therefore a more reasonable way of modelling the river might lead to much lower goal-directedness. And if we put the river in a setting where some other system sometimes disrupted its path, then since the river would not adapt to overcome these disruptions, this would also lead to low goal-directedness.
>
> **The necessity of assuming a causal model.**
> Something we perhaps under-emphasised in the paper is that if you have black box access to the environment or a simulator, then you don’t need an explicit causal model, you just need the ability to measure the variables you’re interested in under various different policies. So that’s one way in which you can avoid worrying about hidden confounders. We have updated the paper to make this clearer.
>
> Of course, we still need to be measuring the important variables. Interpretability-based approaches to measuring goal-directedness may have an advantage over our behavioural approach here. We have added this point to a discussion of the pros and cons of taking a behavourial vs mechanistic approach to measuring goal-directedness in the paper.
>
> The second part of your question is about what happens if an agent is goal-directed with respect to variables we can’t measure because they are internal to the agent. There are two cases here: either the agent is also (perhaps instrumentally) goal-directed with respect to external variables, or it isn’t (as in the case of “wire-heading”). In the first case, we may be able to measure this by considering the external variables. The second case is of less concern to us, since our motivation is about the risks arising from agents pursuing goals in their environment.
>
> Thanks again for the thoughtful review.

---

> > ### Comment · Reviewer_8tgY · 2024-08-10
> >
> > Thank you for the interesting response.
> >
> > "Something we perhaps under-emphasised in the paper is that if you have black box access to the environment or a simulator, then you don’t need an explicit causal model, ..."
> >
> > - I would definitely emphasise this more in the paper. This makes the paper much stronger than my initial impression.
> >
> > I have decided to maintain my score.

---

> > > ### Author Response · Authors · 2024-08-13
> > > **Agreed**
> > >
> > > Thanks. We agree this point makes the paper much stronger and will indeed emphasise it a lot more.

---

### Author Rebuttal · Authors · 2024-08-07

Thank you to the reviewers for their feedback, which has already allowed us to improve the paper. We are pleased that the reviewers agree that  the problem we are tackling is an important one and that our approach is novel.

We have responded to each reviewer’s points individually. Minor presentation issues have all been fixed without comment.

---

### Decision · Program_Chairs · 2024-09-25

**Decision:**

Accept (spotlight)

**Comment:**

The paper makes a significant theoretical contribution to the study of goal-directedness in decision-making systems, with potential implications for AI safety.
However, the work would benefit from addressing the computational limitations, broadening the philosophical discussion, and providing more extensive empirical validation.
Despite these concerns, the reviewers agree that the paper is an important step forward in this research area and recommend acceptance, provided that the identified weaknesses are addressed in the final version